

# Ecosystem connections in the shelf sea environment using complex networks

Ieuan Higgs[1, 2], Jozef Skákala[2, 3], Ross Bannister[1, 2], Alberto Carrassi[1, 4], and Stefano Ciavatta[5]

[1]University of Reading, Meteorology Department UK
[2]National Centre for Earth Observation UK
[3]Plymouth Marine Laboratory UK
[4]Dept of Physics and Astronomy "Augusto Righi", University of Bologna IT
[5]Mercator Ocean International FR

**Correspondence:** Ieuan Higgs (i.higgs@pgr.reading.ac.uk)

**Abstract.** We use complex network theory to better represent and understand the ecosystem connectivity in a shelf-sea environment. The baseline data used for the analysis are obtained from a state-of-the art coupled marine physics-biogeochemistry model simulating the North-West European Shelf (NWES). The complex network built on model outputs is used to identify the functional types of variables behind the biogeochemistry dynamics, suggesting how to simplify our understanding of the

5 complex web of interactions within the shelf-sea ecosystem. We demonstrate that complex networks can be also used to understand spatial ecosystem connectivity, both identifying the (geographically varying) connectivity lengthscales and the clusters of spatial locations that are connected. These clusters indicate geographic regions where there is a substantial flow of information between the degrees of freedom within the ecosystem, while information exchange across the boundaries of these regions is limited. The results of this study help to understand how natural, or antrophogenic, perturbations propagate through the

10 shelf-sea ecosystem, and can be used in multiple future applications such as stochastic noise modelling, data assimilation, or machine learning.



## 1 Introduction

Although shelf seas, understood as the seas covering parts of continental shelf, are only 7% of the global ocean, they are responsible for 20% of the global biological productivity, contribute 20% of the ocean uptake of atmospheric carbon, and are the grounds for 80% of global fish catches (Pauly et al., 2002; Borges et al., 2006; Jahnke, 2010; Legge et al., 2020). For the European economy, the North-West European Shelf (NWES) is of key importance. Numerical models such as the European Regional Seas Ecosystem Model (ERSEM) have been developed in an effort to understand and predict marine ecosystem behavior and the cycling of chemical elements such as carbon, nitrogen, or phosphorus (Heinze and Gehlen, 2013; Butenschön et al., 2016; Ford et al., 2018). However, marine biogeochemistry is complex to simulate, e.g., the ERSEM model contains more than 50 pelagic variables and hundreds of parameters (Butenschön et al., 2016), representing plethora of processes. Such complex model is computationally costly, which makes it unsuitable to address certain questions that require large and long ensemble simulations, such as ecosystem's response to climate change and anthropogenic pressures across large variety of scenarios, or to explore what-if types of analyses for management and policy-making scenarios. Nevertheless, it may be possible to gain insights into such questions through statistical tools leading to innovative representations of complex model outputs, such as those based on network theory (Zanin et al., 2016; Albert and Barabási, 2002), and using them to construct reduced complexity models, such as (but not exclusively) machine learning (ML) emulators (Schartau et al., 2017; Sonnewald et al., 2021).

Networks are a mathematical tool for modelling the key relationships/connections between objects/data. Typically, most networks generated from real-world data are complex networks with examples being found in: biochemical systems; neural networks; social networks; the Internet; and the World Wide Web (Boccaletti et al., 2006). Within the context of environmental networks, and particularly for highly multivariate cases such as biogeochemical marine ecology, networks offer an intuitive human-interpretable view of the highly interconnected spatio-temporal regions and variables (Tsonis et al., 2006) that can often be critical to the resilience of a given system (Barabási and Bonabeau, 2003; Jeong et al., 2001), as well as to better understand how certain pressures or changes in the environment will propagate across the system (Jiang et al., 2018). By using complex networks we gain insight into the structure of the data, the patterns that form, but also an abstraction that will allow for smarter decision-making when considering data sampling and feature selection for ML.

In this work, we use complex networks (CN) and associated statistical analyses, together with NWES as a test case, to investigate three relevant questions related to shelf sea biogeochemistry: (i) We used the network connectivity to estimate the spatial correlation lengthscales of the biogeochemical variables. Such analysis provides important information for the operational marine forecasting systems, if they are based on variational data assimilation (DA) and use parametrised background covariances. This is the case of the UK operational system for the NWES (e.g. (Edwards et al., 2012; Waters et al., 2015; Fowler et al., 2022)) run at the UK Met Office, using ERSEM as the biogeochemical model. Since future observational missions will provide new biogeochemical variables (such as nutrients, or pH) for assimilation (Skákala et al., 2021; Ford, 2021), it is of crucial importance to gain understanding of how transferable the correlation lengthscales are between the different biogeochemical variables. (ii) We exploited clustering network algorithms to demonstrate how a shelf sea can be split into geographic regions,



based on high ecosystem inter-connectivity within the regional boundaries and little beyond them. Because of the significant connectivity in the ecosystem within each identified region, we do expect that ecosystem characteristics will remain similar within the regional boundaries, thus justifying a region-informed modelling strategy. (iii) Finally, we used CNs to identify the local interactions between the modelled biogeochemical variables, subsequently grouping these variables into sets of functional

types. These are also important to select and guide new observational missions.

The analysis from this study can provide additional information to biogeochemistry modellers for building simplified (yet realistic with respect to the objectives) computationally cheaper models than ERSEM. Simultaneously, it can identify the necessary model complexity to simulate realistically the NWES biogeochemistry. Finally, for the goal of developing efficient ML-based emulators of ERSEM, or of some of its critical parametrizations or sub-components, this study paves the path on

how to effectively perform efficient feature selection (i.e., how to select the minimum number of input variables to achieve the desired accuracy).

The paper is organised as follows. We first give, in Sect. 2, details on the model used, explaining each component used to output the data we analysed, as well as the the relevant configurations for each component. In Sect. 3 we discuss the methods used, starting with the preprocessing step used to remove the seasonal signal from the data analysed. We then detail the

approach used to estimate the mean lengthscale of each biogeochemical variable, as well as how we develop a series of spatial networks that help to efficiently capture the spatial variability of these lengthscales. We then explain the clustering algorithm used on these networks that split the shelf sea into a set of regions. The final part of the methodology moves away from the spatial analysis of the variables and gives detail on how we developed a CN to compare the inter-variable interactions and clusters that form. Following this, we present and discuss our results in Sect. 4, each corresponding to a subsection of the

methodology. We finish with concluding remarks, in Sect. 5, summarising the key findings and discussing future work.



## 2 Model and Data

To obtain a complex picture of the shelf sea biogeochemistry, including the relationships between the variety of key biogeochemical variables (detailed in Table 1), the observations are far from sufficient, as any robust observations are limited to only very specific variables i.e., total, or phytoplankton functional type (PFT) surface chlorophyll obtained from the satellite ocean color (Groom et al., 2019). Even those robust observations, exclusively obtained from the satellite have many data gaps and spatially correlated errors, that make their use unsuitable for this type of analysis. Any other observations are typically very rare and extremely sparse (Telszewski et al., 2018), and so provide almost no information on the connectivity between variables and spatial locations across the shelf sea domain. To overcome this limitation to analyse the shelf sea ecosystem connectivity, we used the complex network theory to study the daily NWES surface outputs of a 3-year long (2016-2018) run of the coupled hydrodynamic-biogeochemical ecosystem model NEMO - FABM - ERSEM [1]. The outputs were obtained from a configuration at 7 km grid size, on the Atlantic Margin Model (AMM7) domain (see Fig. 1). The analysis was also repeated on a subset of variables for an independent 3-year long period between 2005-2007 with similar outcome (not shown). The physics and biogeochemistry components of the coupled model that produced those simulations are described below.

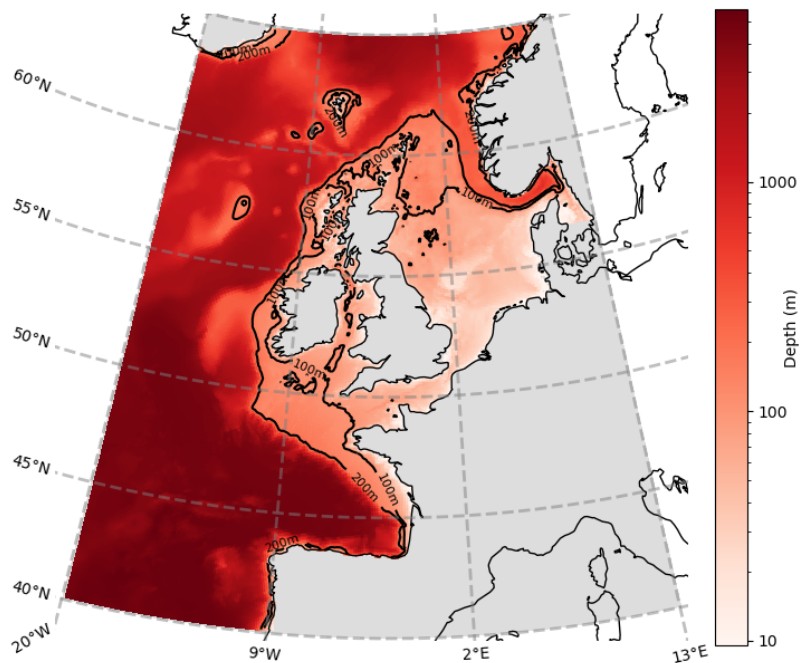

**Figure 1.** The Atlantic Margin Model (AMM7) domain used in this study. The figure also shows the ocean bathymetry of the North West European Shelf

---

[1] NEMO - Nucleus for European Modelling of the Ocean (Madec, 2015), FABM - Framework for Aquatic Biogeochemical Model (Bruggeman and Bolding, 2014)



## 2.1 Physical model: NEMO

The NEMO ocean physics component (OPA) is a finite difference, hydrostatic, primitive equation ocean general circulation model (Madec, 2015). The specific NEMO configuration used in this study has been described in Skakala et al. (2020): it is known as CO6 NEMO, and is based on NEMOv3.6, which is a development of the CO5 configuration described in detail by O'Dea et al. (2017). The model has a spatial resolution of 7 km on the AMM7 domain and employs a terrain following z* - $\sigma$ coordinate system with 51 vertical levels (Siddorn and Furner, 2013). The lateral boundary conditions for physical variables at

the Atlantic boundary were obtained from the outputs of the UK Met Office's 1/12° North Atlantic model (NATL12) (Storkey et al., 2010), while the Baltic boundary values were derived from a reanalysis produced by the Danish Meteorological Institute for CMEMS. The model was forced at the surface using atmospheric fluxes from an hourly and 31 km resolution realization (HRES) of the ERA5 data set (https://www.ecmwf.int/), and utilized annually varying river discharge data from Lenhart et al. (2010).

## 2.2 Biogeochemical model: ERSEM

ERSEM (Baretta et al., 1995; Butenschön et al., 2016) is a marine biogeochemistry model that simulates lower trophic levels of the ocean ecosystem, including plankton and benthic fauna (Blackford, 1997). The ERSEM 50 pelagic state variables are listed in Table 1. The model divides phytoplankton into four functional types based on size: picophytoplankton, nanophytoplankton, microphytoplankton and diatoms (Baretta et al., 1995). ERSEM uses variable stoichiometry for the simulated plankton groups

(Baretta-Bekker et al., 1997; Geider et al., 1997) and represents the biomass of each functional type in terms of chlorophyll, carbon, nitrogen, and phosphorus, with diatoms also being represented by silicon. ERSEM predators consist of three types of zooplankton (mesozooplankton, microzooplankton, and heterotrophic nanoflagellates), with organic material being decomposed by a single type of heterotrophic bacteria (Butenschön et al., 2016). The model represents three different types of detritus and three types of dissolved organic matter (DOM). The inorganic component of ERSEM includes nutrients such as nitrate,

phosphate, silicate, ammonium, and carbon, as well as dissolved oxygen. The carbonate system is also included in the model (Artioli et al., 2012).





**Table 1.** The ERSEM pelagic variables.

| Functional Part of Model | Type | Chemical Components | Abbreviation |
|---|---|---|---|
| Primary Producers - Phytoplankton | Diatoms | Chlorophyll, Carbon, Phosphorus, Nitrogen, Silicon | Diat-Chl, -C, -P, -N, -Si |
| | Microphytoplankton | Chlorophyll, Carbon, Phosphorus, Nitrogen | Micro-Chl, -C, -P, -N |
| | Nanophytoplankton | Chlorophyll, Carbon, Phosphorus, Nitrogen | Nano-Chl, -C, -P, -N |
| | Picophytoplankton | Chlorophyll, Carbon, Phosphorus, Nitrogen | Pico-Chl, -C, -P, -N |
| Predators - Zooplankton | Microzooplankton | Carbon, Phosphorus, Nitrogen | Microzoo-C, -P, -N |
| | Heterotrophic Flagellates | Carbon, Phosphorus, Nitrogen | Flage-C, -P, -N |
| | Mesozooplankton | Carbon | Mesozoo-C |
| Decomposers | Heterotrophic Bacteria | Carbon, Phosphorus, Nitrogen | Bact-C, -P, -N |
| Detritus | Small | Carbon, Phosphorus, Nitrogen | Det:S-C, -P, -N |
| | Medium | Carbon, Phosphorus, Nitrogen, Silicon | Det:M-C, -P, -N, -Si |
| | Large | Carbon, Phosphorus, Nitrogen, Silicon | Det:L-C, -P, -N, -Si |
| Dissolved Organic Matter (DOM) | Labile | Carbon, Phosphorus, Nitrogen | DOM:Lab-C, -P, -N |
| | Semi-labile | Carbon | DOM:Sem-C |
| | Refractory | Carbon | DOM:Ref-C |
| Inorganic Matter | Nitrate | Nitrogen | NO3 |
| | Ammonium | Nitrogen | NH4 |
| | Phosphate | Phosphorus | PO4 |
| | Silicate | Silicon | SiO4 |
| | Oxygen | Oxygen | O2 |
| | Dissolved Inorganic Carbon | Carbon | DIC |
| Other | Bioalkalinity | – | Bio-Alk |





## 3 Methodology

### 3.1 Data Preproccesing

In order to extract non-trivial interactions and dynamics of the system, we removed the dominating seasonal signal. Typically,
this is achieved by phase-averaging and standardising the data to generate an anomaly time series (with respect to climatology) with zero mean and unit variance. However, with our high temporal resolution (daily) but just 3 years of data, this phase-averaging method can heavily skew a dataset with both high inter-annual and daily variability. As a result, we instead opted to use a high-pass filter that standardises every time step of data according to its local temporal behaviour, with details given in the following.

First, for each day, we computed the local mean that bears the signature of the seasonality. This is done by averaging the values within a chosen time-window centered on that day:

$$\mu_d = \frac{1}{T} \sum_{t=-T/2}^{T/2} a_{d+t}, \tag{1}$$

where $T+1$ is the number of days in the window, with $T$ being an even natural number, and $a_{d+t}$ represents the value in the time series at day $d + t$, with offset (in days) $t$. The aim is to remove the seasonal cycle by subtracting $\mu_d$, i.e. $a_d - \mu_d$. Selecting an
appropriate window size is crucial in generating a time series with useful properties. For our purpose of removing the seasonal cycle, a relatively short window is appropriate. Following a sensitivity analysis (not shown) we found that a window of $T = 10$ days efficiently removes the seasonal effect while retaining a functional signal for further analysis. We then take the standard deviation of those same points within the window:

$$\sigma_d = \sqrt{\frac{1}{T-1} \sum_{t=-T/2}^{T/2} (a_{d+t} - \mu_d)^2}. \tag{2}$$

Using the output from both Eqs. (1) and (2) we transform the raw data points into a time-local standardised form:

$$a_d^n = \frac{a_d - \mu_d}{\sigma_d}. \tag{3}$$

The data consists of 50 ERSEM state variables (as well as temperature and salinity), on a $375 \times 297$ horizontal grid using only the surface layer with 1094 days ($> 6$ billion data points). Following the preprocessing stage, the data will have a mean of $0$ and unit variance relative each point's surrounding temporal behaviour, as set by window size of 10. Unless explicitly specified,
all data used in this study have been preprocessed using the procedure in Eqs. (1), (2) and (3).

As hinted to above, our method is designed to mitigate (or possibly remove) the skewness caused by instances of high inter-annual variability (those would eventually not be an issue when working with dataset on a much longer period than 3 years).





Furthermore, the proposed approach is still effective in removing the seasonal cycle from data and it is more sensitive than phase-averaging to dynamics in both low- and high-activity periods of the time series. A key limitation is that the method is

not suitable if we had intended to compare data points and times that are separated by an offset significantly larger than $T$.

## 3.2    Horizontal lengthscale estimates

### 3.2.1    Biogeochemical Lengthscale Estimation

To estimate the lengthscale of each biogeochemical variable, we calculated a Spearman's correlation between the time series at a reference point and all of its surrounding points simultaneously, within a selected radius. This should reduce the number

of unnecessary computations, while still being confidently large enough to capture the lengthscales. We intentionally chose to use the Spearman correlation in order to capture non-linear relation that would have been otherwise masked with the Pearson correlation. Starting with a circle of small radius (7 km in this case), we calculated the mean correlation of grid points within the circle. If this correlation was above some threshold, we increased the radius of the circle and recalculate the new mean. Once the mean correlation dropped below a given threshold, we stopped increasing the radius and took this to be an estimate

of the lengthscale at the given threshold. The exact distance calculated for these lengthscales changes according to the selected correlation threshold (increasing as the selected threshold decreases). A selection of appropriate thresholds (0.5, 0.6 and 0.7) were chosen to capture the lengthscales that demonstrate a "strong" correlation. While this approach is relatively simple, it provides a quantitative measure of the lengthscale, and is easily be applied to each variable in the system. This does assume the ocean is homogeneous and isotropic that, while not necessarily true, is a useful simplification.

Figure 2 provides an example for calculating the lengthscale of Diatoms Chlorophyll with a correlation threshold of 0.5 in the centre of the North Sea.



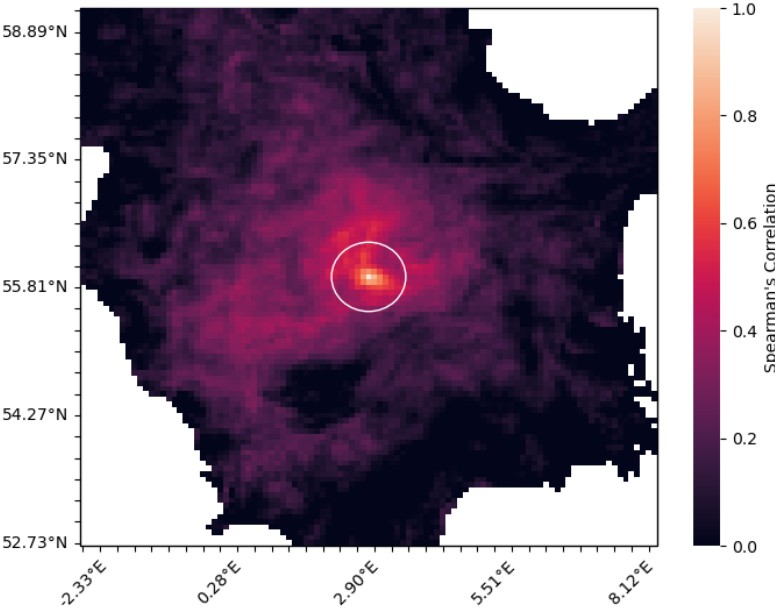

**Figure 2.** Lengthscale estimate, $\ell = 45.5$ km shown as a white circle, calculated for Diat-Chl with correlation threshold set at 0.5, for the reference point with coordinates $56.10°N, 3.20°E$. The colours represent the simultaneous Spearman correlation of each grid point with the central grid point (which is of value 1, as it perfectly correlates with itself).

We averaged these lengthscales over the domain for each variable, and were tested at multiple correlation thresholds (0.5, 0.6 and 0.7). These lengthscales are used later in Sect. 4.1.

### 3.2.2 Generating spatial networks

The method used to construct spatial networks from the biogeochemical model is largely inspired by similar applications to models of the climate system (Tsonis et al., 2006). We considered each grid point to be a node in a network for a given variable. In order to generate the links between these nodes, we calculated a Spearman's correlation coefficient between the time series of every pair of nodes for the same variable. To make these comparisons computationally feasible, the grid points are upscaled to a 21 km spatial resolution. We then pruned this network by removing the links with the weakest correlation, although, as

we shall clarify later, the exact pruning method was dependent on the task. The first method (used in Sect. 3.2.3) required dynamically adjusting the cutoff threshold for the spatial network of each variable so that each network contained the same number of nodes and links - meaning that the only difference between the network of any given variable to another is found in its structure. As such, any lengthscales of the same point (but different variables) derived from these networks can be directly compared to each other. The second method (used in Sect. 3.3) consisted of picking a suitable cutoff correlation threshold,

whereby a link was ignored if the correlation was beneath a chosen value.





### 3.2.3 Estimating horizontal lengthscale from the spatial networks

As opposed to the biogeochemical lengthscales computed in Sect. 3.2.1, which refer to each variable and reflect their physical properties averaged on the domain, here we manipulate the spatial networks to look at the spatial dependency of this lengthscale. A lengthscale is calculated for every node in a dynamically thresholded horizontal connectivity network (as desribed

in Sect. 3.2.2), meaning we defined a unique lengthscale at every point for each variable. Figure 3 shows the method used to calculate the lengthscale estimation for a given node in a single network (shown as a black dot in the centre of each panel). Figure 3a shows the set of nodes connected to the current target node (red), indicating the surrounding grid points that correlate strongly through time with the current target node. Figure 3b highlights the area represented by this connected set of nodes. Note that the shape of this area is often irregular, but we can generally expect that nodes closer to the target node have a higher

chance of being connected. Figure 3c shows a circle with equivalent area to the area highlighted in Fig. 3b. We took the radius of this circle as an approximation of the horizontal lengthscale for this node. The radius of this circle is used as an approximation of the horizontal lengthscale for this node. We assume isotropy and that strong connectivity exists only within this radius, as implied by the transition between Fig. 3a and 3c. This approximation is a reasonable assumption for the purpose of our analysis as it allows us to make direct comparisons between nodes and across variables, capturing clear and scale-relevant

features of the domain. However, it should be noted that these correlations are inherently anisotropic, as we shall see in the regionalisation results in Sect. 4.3.

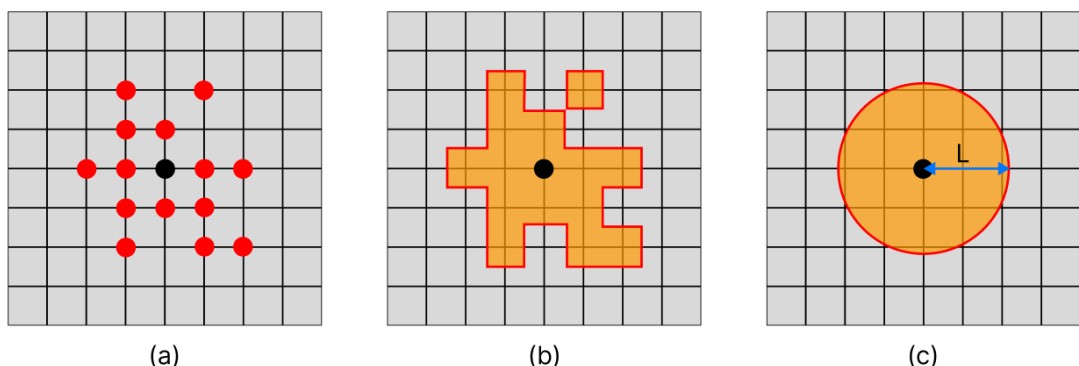

**Figure 3.** Method for calculating the lengthscale of a given node in a network representing the horizontal connectivity of a variable. Panel (a) shows the target node (black) and each of the nodes on the grid that it is linked with (red). Panel (b) shows the area represented by each of the nodes linked to the current target node. Panel (c) shows a circle with equivalent area to the area covered in Panel (b), and the radius $L$, provides us with our approximation of the lengthscale at this point.

With the spatial variation captured for each variable, we sought to find any underlying structure that was shared between variables across the surface layer. This was done by taking the mean lengthscale at each grid point across all variables from the dynamically thresholded spatial networks. In order to assess whether this spatial variation could be well approximated





by the mean of these lengthscales, we compared the spatial distribution of lengthscales between each different variable using Pearson's correlation. Here, we would expect to see a high correlation if the structure of the spatially varying lengthscales is consistent. This set of spatially varying lengthscales was then represented as a ratio of the mean.

### 3.3 Regionalisation using spectral graph clustering

With the spatial networks, the *graphs*, from Sect. 3.2 at hand, we can manipulate them to extract information of interest.
We used *spectral graph clustering* (SGC), whereby the spectral properties of key network matrices are considered instead of working on the data directly. In particular, with SGC we partitioned the network into clusters by utilizing the eigenvectors of the Laplacian matrix. This makes it possible to apply the clustering algorithm to "global objects" (the eigenvectors) of the full dataset under study, as opposed to standard applications of clustering methods to the individual entries of the dataset.

For a network with $n$ nodes the graph Laplacian matrix is a square matrix $\mathbf{L} \in \mathbb{R}^{n \times n}$, which can be intended as a discrete
(for network) counterpart of the classical Laplacian operator for continuous variables, measuring in this case how the strength of a node changes in its surroundings. For finite dimensional networks, such as those constructed in this work, $\mathbf{L}$ is obtained as the difference between the *degree matrix*, $\mathbf{D} \in \mathbb{R}^{n \times n}$ and the *weighted adjacency matrix*, $\mathbf{W} \in \mathbb{R}^{n \times n}$ of the network, $\mathbf{L} = \mathbf{D} - \mathbf{W}$. Here, $\mathbf{W}$ represents the weight of an link between nodes (as defined in this case by the Spearman correlation), $\mathbf{D}$ is a diagonal matrix that represents the sum of the weights that each node has to other nodes. A node with a large degree
results in a large diagonal entry in the Laplacian matrix, which may dominate the properties of the matrix. To address this, the Laplacian matrix is normalised to make the influence of these nodes more similar to other lesser connected nodes, giving $\mathbf{L}_{\mathrm{sym}} := \mathbf{D}^{-1/2} \mathbf{L} \mathbf{D}^{-1/2} = \mathbf{I} - \mathbf{D}^{-1/2} \mathbf{W} \mathbf{D}^{-1/2}$. Thanks to this normalisation, we can apply a static threshold to each of the networks (as mentioned in Sect. 3.2.2). Once the desired number of clusters, $k$, is chosen we compute the first $k$ eigenvectors $u_1, ..., u_k$ of $\mathbf{L}_{\mathrm{sym}}$ and arrange them as columns of matrix $\mathbf{U} \in \mathbb{R}^{n \times k}$. We then form the matrix $\mathbf{T} \in \mathbb{R}^{n \times k}$ by normalising
the rows of $\mathbf{U}$ such that $t_{i,j} = u_{i,j}/(\sum_k u_{ik}^2)^{1/2}$. For $i = 1, ..., n$, let $\mathbf{y}_i \in \mathbb{R}^k$ be a vector corresponding to the $i-th$ row of $\mathbf{T}$. Using k-means, we collect the vectors $(\mathbf{y}_i)_{i=1,...,n}$ into clusters $C_1, ..., C_k$. Finally, we output clusters $A_1, ..., A_k$ with $A_i = \{j | \mathbf{y}_j \in C_i\}$, mapping back from the eigenspace and providing a cluster label for each node in the network (Ng et al., 2001; Von Luxburg, 2007).

The method has several characteristics that make it preferable to other clustering methods that are applied to the dataset
directly (i.e., k-means, hierachical clustering, DBSCAN). The key advantages of SGC that made it ideal for our purposes are the following. (i) Handling non-linearly separable data. Since the eigenvectors of the graph Laplacian capture the structure of the network, they can provide a useful representation of the data even if it is not linearly separable. (ii) Being strongly robust to noise. Since the eigenvectors are computed using the entire network, they are less sensitive to noise and outliers compared to traditional clustering methods that rely on individual data points. (iii) Identifying clusters of different shapes and sizes.
Traditional clustering methods such as k-means are limited to identifying spherical clusters of similar size.

Nevertheless, a key challenge in SGC is selecting the appropriate number of clusters to use with the algorithm. A common solution to this problem is to use the "eigengap heuristic" (Tibshirani et al., 2001), which uses the amplitudes and rate of change of the eigenvalues of $\mathbf{L}_{\mathrm{sym}}$ to identify the optimal number of clusters. The method suggests that if the difference between the



$k-th$ and $k+1-th$ eigenvalues (an eigengap) is substantial, $k$ is more likely to produce a correct number of clusters. However,
if the gap is small, it may lead to less reliable clustering results as perturbations may cause the eigenvectors to be swapped. Our
preliminary analysis (not shown) overall indicated no obvious choice for the cluster number, with no significant eigengaps. To
address this, we opted to apply the clustering algorithm with every cluster number $1 \leq k \leq 20$ (an upper bounded selected for
computational affordability). This allowed us to evaluate the quality of the clusters at different cluster numbers, identifying the
values that produce the clearest patterns and structures.

In order to compare the regionalisation of each variable, we first projected the cluster labels of each node back onto the
horizontal plane. Then, we applied an edge detection kernel to identify the boundaries between differently labelled regions,
creating a boundary map for each variable (with value 1 at boundary grid points and 0 elsewhere). Then, in order to assess how
"robust" any given boundary is across the set of variables, we calculate the fraction of variables that have a boundary, at each
grid point.

## 3.4 Inter-variable interaction networks

With the work in the previous sections being focused on understanding how each variable separately behaves in horizontal
space, we then focused on developing an understanding of how the variables interact with each other co-spatially. This was
achieved by assessing the interactions between the different biogeochemical state variables of ERSEM, computing the abso-
lute value of Spearman's rank correlation coefficient between the time series of each variable at an ERSEM grid point. As
with before, we chose Spearman's correlation to capture any potential non-linear, monotonic links between variables. These
correlations can be represented as weighted adjacency matrix, where the rows and columns represent each of the variables, and
each matrix's entries represents the strength of a pairwise connection between variables at a grid point. As one might expect,
the strength of these correlation coefficients will vary spatially. Therefore, in order to identify the most consistent and robust
connections (and in a computationally efficient way), we calculated an adjacency matrix for 300 points randomly sampled
across the shelf (bathymetry $\leq 200$ m). These 300 matrices are then averaged so that each entry represents the mean of each
pair-wise comparison. To ensure that these averaged values are reliable, we then calculated the coefficient of variation,

$$\text{CV}_{\text{i,j}} = \frac{\sigma_{i,j}}{\mu_{i,j}}, \tag{4}$$

where $\mu_{i,j}$ is the mean of correlation coefficients (across all 300 sample points) between the time series of variables $i$ and
$j$, and $\sigma_{i,j}$ is the standard deviation of the same set of coefficients. This coefficient is then given as a percentage so we can
intuitively view the standard deviation of the correlations as a percentage of the mean.

We accounted for any processes that occur on a lagged or delayed time scale through cross-correlation - determining the
degree to which one time series is correlated with another time series after shifting the latter series forward or backward in time.
The correlation between any variable pair in the results are always shifted by an offset that maximises the correlation between
the two variables. It should be noted however that, as a result of the pre-processing step applied to the data (cf Sect. 3.1), the





time offset is bound to $T$; by construction (after the pre-processing) points in a time series that are outside of that range are insensitive to each other.

As this inter-variable analysis provided us with a weighted adjacency matrix, we were once again able to apply the SGC algorithm described in Sect. 3.3. In this instance, the resulting clusters can be interpreted as functional groups of similarly behaving variables, rather spatial regions of similar behaviour.

## 4    Results and Discussion

### 4.1    Horizontal lengthscale estimates

Figure 4 shows the estimated correlation lengthscales for each model variable using three correlation thresholds (0.5, 0.6 and 0.7) as found from the analysis described in Sect. 3.2.1. Lengthscales appear significantly different across the set of ERSEM variables, with salinity notably longer than the ERSEM biogeochemical variables. This has profound implications for includ-

ing new types of biogeochemistry observations into the operational model of the NWES. The present version of the NWES operational system uses parametrized correlation lengthscales in its DA scheme (Waters et al., 2015; Skákala et al., 2018; Skakala et al., 2020; Skákala et al., 2021; Fowler et al., 2022) . Therefore, any new assimilated variables require prior knowledge of those lengthscales. The variability in Fig. 4 implies that the lengthscales of any new assimilated variables cannot be simply deduced from the known lengthscales of established assimilated biogeochemical, or physical variables, such as surface

chlorophyll, or sea surface temperature. This is particularly relevant given that new missions based on automated observing platforms, such as gliders (Telszewski et al., 2018), are starting to deliver data for assimilation for a much broader class of variables than we were used to. For example, assimilation of oxygen measurements from gliders was recently introduced by assuming similarity between the lengthscales of oxygen and chlorophyll (Skákala et al., 2021). As seen in Fig. 4, this assumption is not found to be true, since the chlorophyll component for each phytoplankton functional type (Diat-Chl, Micro-Chl,

Pico-Chl and Nano-Chl) and oxygen do not have similar lengthscales, at least according to the model data at the surface. It is expected that in the immediate future assimilation capability for new glider observed variables, such as nitrate, phosphate, or pH, will be included into the system. Our methods can provide variable-specific lengthscale values to be used in the operational system for the assimilation of those new observations.





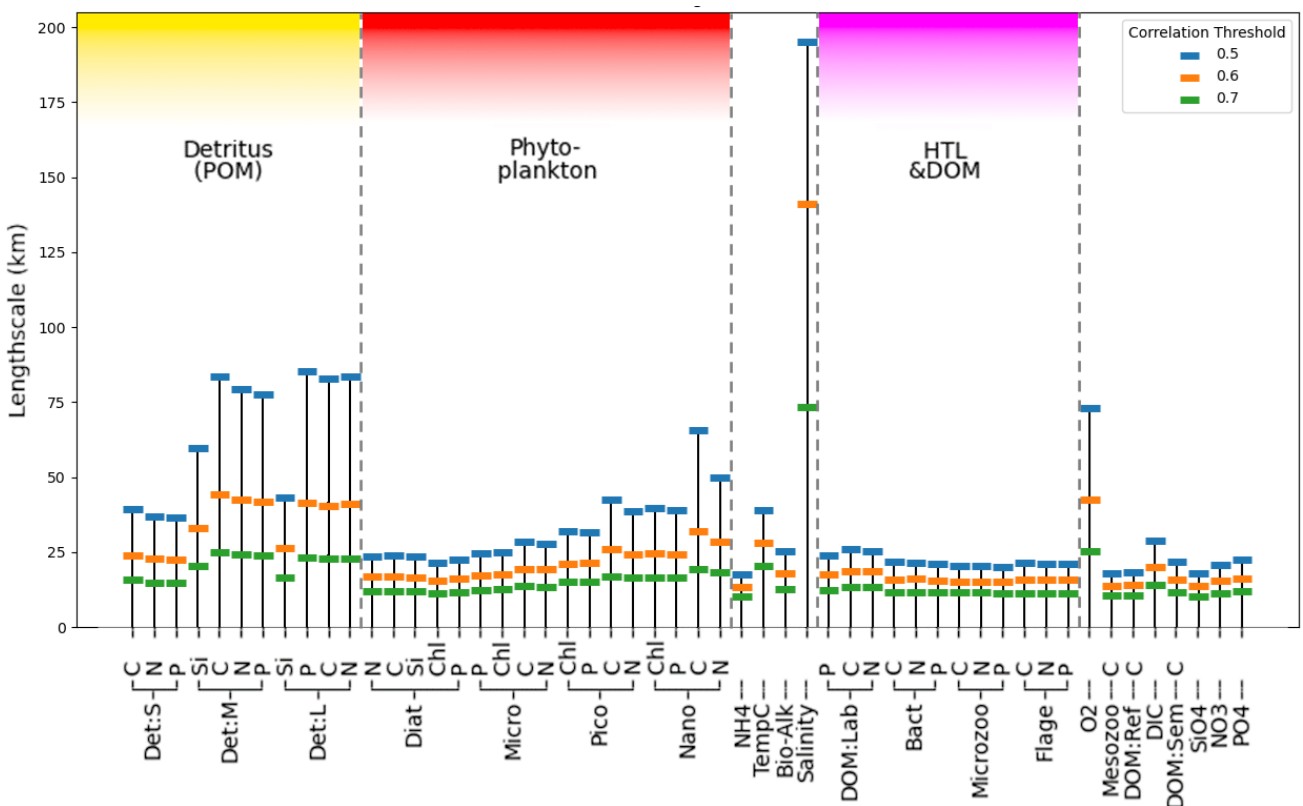

**Figure 4.** Estimate of horizontal lengthscales on for each ERSEM variable (as shown in Fig. 2 for diatoms chlorophyll with threshold 0.5), using correlation metrics to determine distance at which mean correlation drops below thresholds of 0.5, 0.6 and 0.7.

Not only is there a large a variation in the magnitude of mean lengthscales between variables, there is also a large variation in lengthscales spatially. This is examined in Fig. 5 which shows that, as long as a specific scaling of each variable is taken into account, there is a clear, consistent structure to the way in which the lengthscales vary horizontally. In particular, Fig. 5a shows the mean lengthscale across all variables at each spatial point - as derived from dynamically thresholded networks that contain the same number of links and nodes (but differing correlation strengths). These lengthscales are well represented as a mean because the spatial variation in the horizontal lengthscales (Fig. 5b) are generally highly correlated ($> 0.7$) between variables. This indicates that the lengthscale spatial variation is similar across the variables, up to a constant scaling factor effectively shown in Fig. 4. Results in Fig. 4 and Fig 5) suggest that if we write lengthscale $\ell$ as a function of a biogeochemical variable $v$ and spatial location $x$, $\ell(v,x)$, it can be approximately factorized as a product of two independent functions $\bar{\ell}(v), f(x)$: $\ell(v,x) \approx \bar{\ell}(v) \cdot f(x)$. This hugely simplifies the task of attributing lengthscales to new assimilated variables, as the lengthscales can be determined from two independent functions, a variable-dependent mean length $\bar{\ell}(v)$ (Fig. 4) and a dimensionless spatial lengthscale variability function $f(x)$ (Fig.5a). It is worth noting that temperature is the poorest fit to this model, owing to its





weak correlation in Fig. 5b, further suggesting that the lengthscales of the biogeochemical variables cannot be easily derived from those of the physical variables.

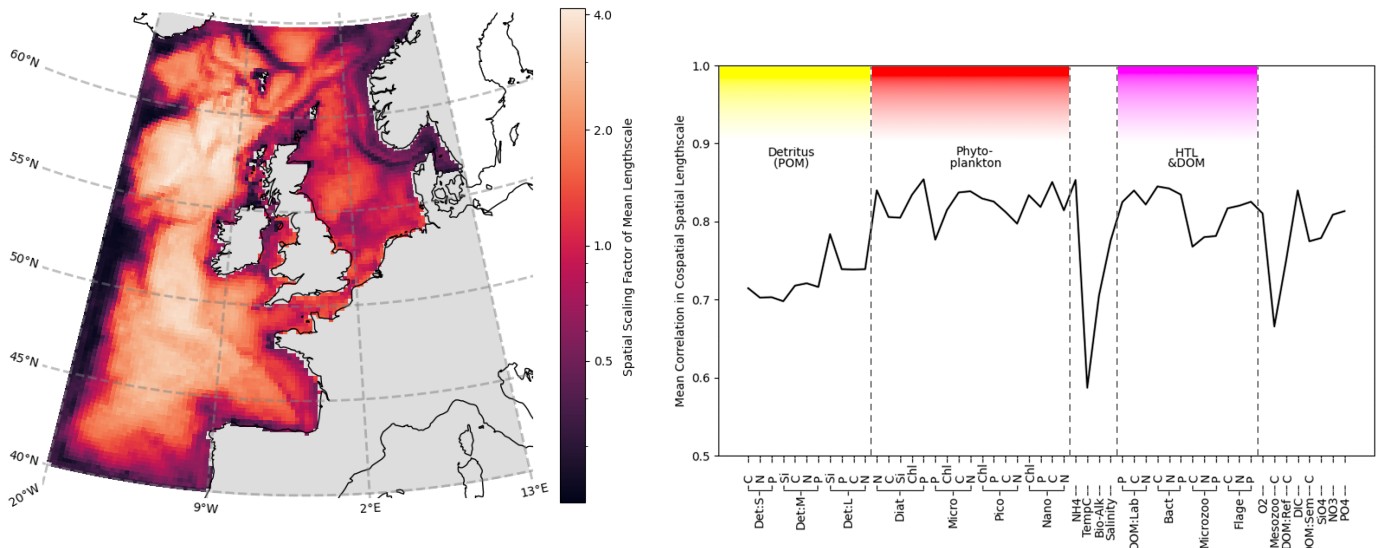

(a) Mean surface lengthscale estimates. Average is taken over the full set of ERSEM variables. Correlation threshold is dynamically adjusted for each variable as described in Sect. 3.2.3.

(b) Mean Pearson's correlation between co-spatial horizontal lengthscales of each variable to every other variable.

**Figure 5.** Horizontal lengthscales vary spatially across ERSEM variables, using correlation network connectivity to approximate the scaling factor. The spatial variation is consistent between different variables, as shown by the co-spatial Pearson's correlation between each variable.

Beyond the applications in DA, these results are relevant also for the design of appropriate strategies for probabilistic pre-
diction or model error compensation. For instance, when considering how to model stochastic noise across the spatial domain, we can clearly see that simply applying white noise across each grid point would be unrealistic, as there are significant spatial correlations to consider (e.g., applying such noise to and initialization to introduce uncertainty in initial value conditions). As outlined, these spatial correlations will also vary in size, meaning that the correlated noise model should be scaled differently according to the target variable.

Of particular interest in Fig. 5 is that whilst on the majority of the domain the variables are spatially-correlated, there are distinctive boundaries, or "cuts" (lines of low-connectivity) between areas in the domain, where the lengthscale decreases rapidly relative to the surrounding areas. The clearest examples of this can be seen along the Norwegian Trench, and in the seas north of Scotland. This indicates that there are lines of extremely low connectivity, where information is not shared across a given boundary, and instead form their own quasi-self-governing community of behaviour. We shall explore this idea further
in Sect. 4.3. It is notable that another area of low-connectivity is the open (Atlantic) AMM7 domain boundary regions. This



indicates that the boundary conditions of the regional model de-correlate from the rest of the domain. The lack of connection between the boundaries and the rest of the domain can be seen as desirable, considering the large uncertainty in the open boundaries of regional ecosystem models. Because of their uncertainty, we have excluded the boundary regions from any further analyses presented here.





## 4.2 Regionalisation using spectral graph clustering

Figure 6 shows the aggregated community boundaries resulting from the use of spectral graph clustering (SGC; see Sect. 3.3) on the spatial networks of each variable. Every point on the map is coloured according to the number of boundaries that pass through it - meaning the brighter, "robust" boundaries are common to the vast majority ($> 75\%$) of variables. Conversely, the darker regions indicate areas where fewer (if any) boundaries exist, across all the variables. From Fig. 6a, at a cluster count of only $k = 3$ (see Sect. 3.3), distinct boundaries form across the Atlantic ocean, and the opening to the English Channel and Irish Sea. These lower-order, larger scales, approximations of the non-shelf boundaries areas are consistent with well-known regionalisations such as the Longhurst Provinces (Longhurst et al., 1995). We focused then on smaller scale shelf sea areas. As the number of clusters chosen increases and the domain splits further, the cluster boundaries on the shelf converge onto more intricately shaped and sized real features. This is illustrated in Fig. 6b for the case with $k = 13$ clusters. At these higher cluster counts, it is clear that the variables share large boundaries, particularly on the shelf. This indicates that they are likely a robust feature of the domain (in Fig. 6 they always represent $75 - 100\%$ of variables). It is important to emphasise that this wouldn't necessarily be the case, as the network construction and regionalisation for each variable are independent of each other, which does show through in the form of some less robust boundaries ($< 40\%$ shared) that appear to be subdividing the more robust regions on the shelf.

We used those robust boundaries to identify 13 regions representing areas of NWES connectivity. Results of this regionalisation are represented in Fig. 7. It is anticipated that between the thirteen regions, each identifies areas with similar biogeochemical/ecosystem characteristic within its boundaries. The converse is not necessarily true: two dynamically disconnected regions that communicate little between each other can still have similar characteristics. However, many of the regional boundaries shown in Fig. 6 and Fig. 7 clearly match the well known geographic features of the area, e.g., German Bight, Southern Bight, Doggerbank, Norwegian Trench, or more broadly, English Channel and North Sea. Our complex network clustering provides also reasonably similar results to the expert-based partition of NWES applied in many works (Ostle et al., 2016; Legge et al., 2020; Fowler et al., 2022), with some extra fine details and some new regions included.





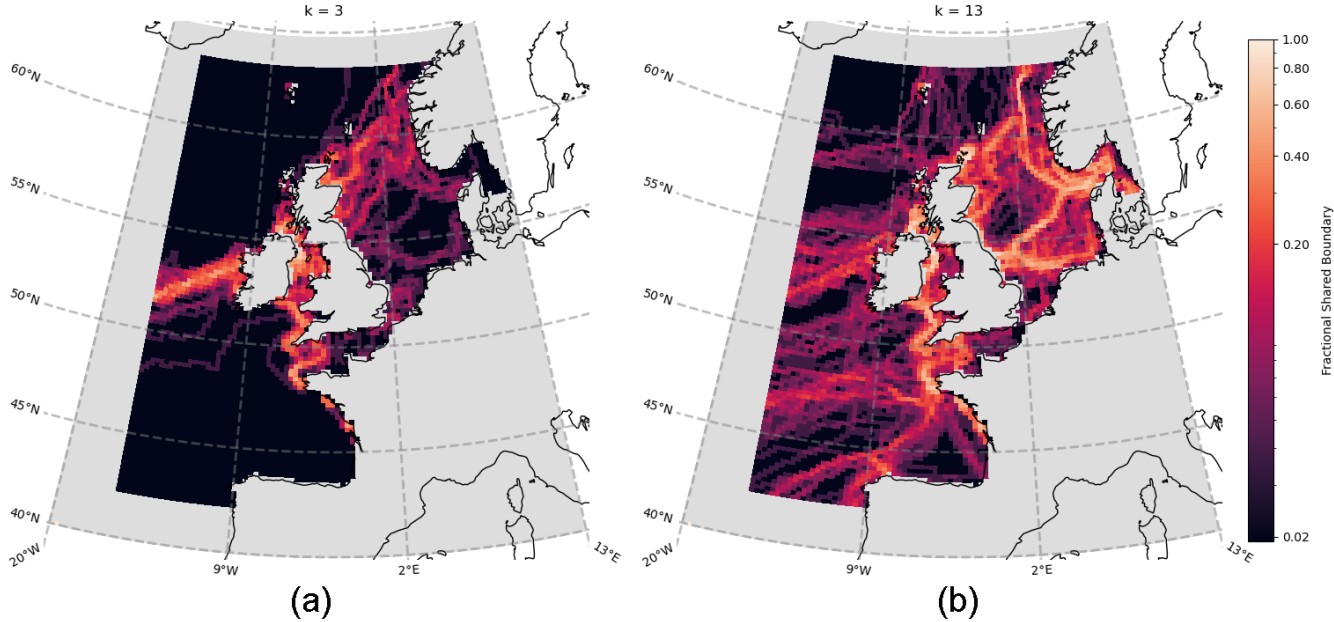

**Figure 6.** Aggregated boundary heatmap generated from community detection (clustering) of the spatial network for each of the 50 ERSEM variables (cf Table 1). Points are coloured according to the number of boundaries (as derived from different variables) that intersect, with bright regions indicating that many variables share a boundary, and darker regions indicating the absence of many boundaries. Panel a shows cluster number $k = 3$, while Panel b $k = 13$.

While we see that the features from Fig. 6 are at least partly driven the bathymetry of the domain (see Fig. 1), the boundaries

particularly seem to reflect shallower bathymetry (approx. $100\,m$) than the $200\,m$ depth usually applied to delimit the margins of shelf-seas, including NWES. As a consequence of that, a large section of the NWES near the open ocean boundary (e.g., Celtic Sea, the north-west section of the NWES) is in fact connected to the open ocean and can be seen as an area of robust shelf-open ocean exchange. Some regional boundaries reflect the properties of the water column often linked to bathymetry, e.g., the boundaries between regions J and K (Fig. 7) in the western English Channel, and similarly the boundary of region

J at the Southern Bight, correspond to the boundary between the permanently mixed and seasonally/intermittently stratified waters (Ostle et al., 2016). The regions C, D along the Norwegian trench and G in German Bight from Fig. 7 are coastal areas influenced by major riverine outflow. The boundaries of those regions effectively delimit the area into which the nutrient and fresh-water rich outflow propagates, as a function of local dominant currents. This can be seen in some more detailed plots of chosen ecosystem indicators across the 13 regions from Fig. 7, which can be found in Fig.S1 of Supporting Information (SI).





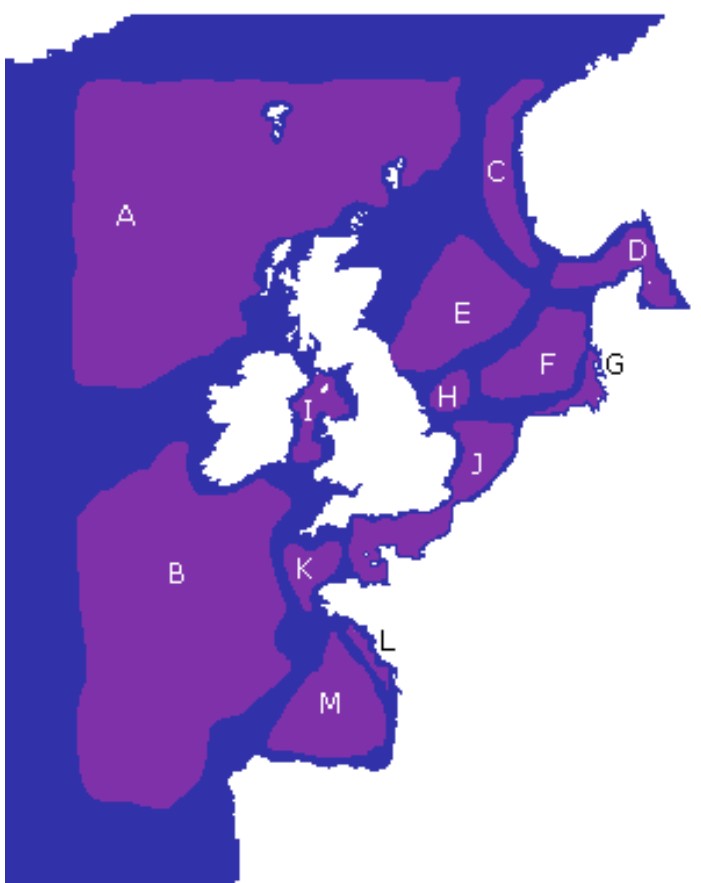

**Figure 7.** The 13 key regions drawn from the regionalisation found in Fig. 6, labelled A-M (purple). Any area not assigned to one of these regions (blue) is due to uncertainty resulting from either domain boundaries or region boundaries.

### 4.3 Inter-variable interaction networks

Figure 8 shows sets of ERSEM state variables that behave functionally as groups, together with a confidence measure to ensure these groups are robust and consistent. More specifically, Fig 8a shows mean absolute values of co-spatial Spearman cross-correlations between each pair of biogeochemical variables at 300 sample points on the shelf (bathymetry $\leq 200\ m$). The diagonal values represent correlations of each variable with itself and are by definition equal to 1. A *hierarchical clustering algorithm* is applied to this matrix to arrange the variable order, creating groups of similar variables that can be easily identified as distinct "blocks" of elements that form around the diagonal (Müllner, 2011). Figure 8b shows the corresponding coefficient of variance, Eq. (4), calculated from the same 300 sample points.

If two variables display a high mean correlation and low coefficient of variation, it indicates that there is a reliable and consistent connection between them in the NWES model data set. The pairwise elements of each group within a high-correlation



"block" tend to show a low coefficient of variation in the corresponding plot, indicating these variables can be grouped together both reliably and consistently.

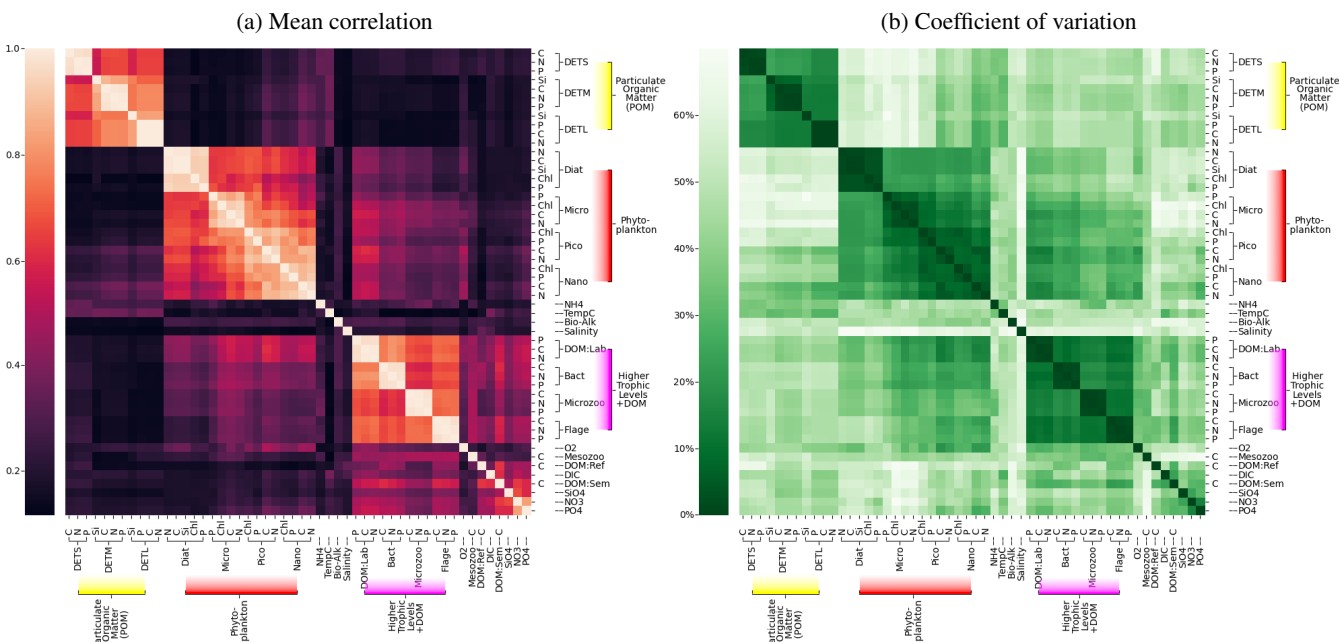

**Figure 8.** Grouping of ERSEM state variables calculated from from 300 sample points on the NWES. Plot (a) shows the mean absolute Spearman correlation between each pair of variables, with the rows and columns sorted using hierarchical clustering, as denoted by the dendrogram. Panel (b) shows the coefficient of variation (Eq. 4), but shown as a percentage, using the same ordering found for plot (a).

Although some of these links among variables could have been anticipated to some degree, the quantitative grouping demon-strates the opportunity, and provides the metric, for researchers aiming to either reduce the complexity of the ERSEM ecosys-

tem model, or build simpler models than ERSEM. For example, Fig. 8a shows that different biomass components (chlorophyll, carbon, nitrogen, phosphorus and silicon) of the same phytoplankton type are strongly correlated and grouped together, which means their values can in principle be reasonably predicted from each other. This implies that despite of the sources of po-tentially significant variability in the phytoplankton stoichiometry, (e.g., variations in its chlorophyll to carbon ratios due to changes in environmental light conditions), Fig. 8 indicates that these are relatively secondary compared to the overall dy-

namics (growth, grazing, mortality, sinking) governing the phytoplankton biomass. As a result of this, we argue that a simpler model can be formulated by fitting the (potentially non-linear) relationships between the different phytoplankton biomass com-ponents and grouping them only into one functional variable. Similarly, another simplification suggested by the connectivity analysis from Fig. 8, is to merge different phytoplankton functional types into one, i.e., mostly the two nanophytoplankton





and picophytoplankton species into a single, "small-size phytoplankton" functional type, along with the larger diatoms and
microphytoplankton groups. Given that the size of the phytoplankton species is known to strongly correlate with various as-
pects of their dynamics, such as photosynthetic light absorption, metabolic and sinking rates (e.g., the smaller species are more
representative of phytoplankton in the open ocean, or oligotrophic areas), such grouping arguably reflect the underlying driving
physical processes. Besides the phytoplankton functional group, Fig. 8 demonstrates two more clusters of variables grouped
together: the group of particulate organic matter (POM), and the group consisting of heterotrophic flagellates, microzooplank-
ton, heterotrophic bacteria and dissolved organic matter (DOM). The links in the latter group are provided through plankton
feeding and excreting organic matter. The groups in Fig. 8 correspond well with the ERSEM model functional type diagram
(Butenschon et al., 2015) (cf Table 1), however their specifics, like the concrete grouping of living and non-living organic
matter, are still quite non-trivial.

Finally, we would like to caution against over-interpreting the Spearman cross-correlation matrix from Fig. 8. The matrix
shows potentially non-linear connections between variables that could be parameterized, reducing the number of model state
variables and replacing a computationally expensive part of the model with cheaper formulations. However, little correlation
in the matrix certainly does not imply little dynamical relationship in the model. For example, phytoplankton and nutrients
are very strongly dynamically interconnected, but their connection is complex and cannot be expressed as a simple monotonic
function captured by the correlation. This demonstrates itself by different signatures of correlation coefficient at different times,
e.g., at times when nutrients are a limiting factor to phytoplankton growth, phytoplankton and nutrients will be positively
correlated, whereas if the nutrients are reduced through uptake, they will be negatively correlated with phytoplankton.

Figure 8 can be naturally interpreted in the context of complex networks analysis, by considering panel (a) as an adjacency
matrix of a network. In this instance, the nodes are the variables and the links are determined by top 25% most robust correla-
tions. To identify the functional groups behind the biogeochemistry variables, we applied SGC (as described in Sect. 3.3 and
Sect. 3.4) to the cross-correlation matrix from Fig.8a and the results are displayed in Fig. 9. Three main groups identified from
the correlation matrix are clearly visible: particulate organic matter (yellow), phytoplankton (red) and the cluster of higher
trophic level variables and DOM (pink). The blue cluster consists of nitrates, phosphates and silicates (the nutrients), and is
only weakly connected to the rest of the network. As already mentioned, being "weakly connected" does not necessarily imply
lack of dynamical connection, it may mean the connection is too complex to be parameterised by a monotonic function. The
observed links in Fig. 9 between the nutrients (nitrates, phosphates, silicates) can be explained through the observed links be-
tween the different plankton/POM/DOM chemical elements represented by the model (carbon, nitrogen, phosphorus, silicon).
These links suggest there is a high degree of connection between the different element cycles. The connection between element
cycles then naturally implies connection between the elements also in their inorganic (nutrient) form. However, there is addi-
tional complexity: the inorganic form of nitrogen is represented in the model by both ammonium and nitrates, which are related
through the process of nitrification. Ammonium dynamics are relatively more complex than the ones of nitrate. Ammonium is
not involved only in phytoplankton assimilation and nitrification, as nitrate - but also in phytoplankton excretion and reminer-
alization of organic matter by bacteria. Ammonium also drives changes in bioalkalinity. This additional complexity might be
behind the fact that ammonium is not part of the "nutrients" cluster, but is separated from all the other variables (Fig. 9).



The cyan cluster consists of temperature, dissolved inorganic carbon (DIC) and semi-labile organic matter, with the dissolved
inorganic carbon being weakly connected to the higher trophic level-DOM cluster. Connections between temperature and gases,
such as $CO_2$, which form the majority of the DIC, are explained by the fact that temperature drives gas solubility in the water.
However, such connections do not always happen on the time-scale relevant to this analysis, e.g. oxygen is not found to be
part of the cyan cluster. This can in part be explained by the longer ($\sim$ 2-3 week) time-scale on which SST drives near-surface
oxygen on the NWES (Skákala et al. (2023)). Crucially, on the shorter time-scales analysed in this study, oxygen is more
strongly linked to the phytoplankton group (Fig. 9) through photosynthesis and respiration. Finally, there are several variables
(e.g salinity, or as we previously mentioned, ammonium) colored in grey that are completely disconnected from the rest of the
network.



**Figure 9.** A network derived from the correlation measures found in Figure 8, where we treat the pair-wise correlations as an adjacency matrix, and apply a spectral clustering algorithm to partition the variable set into functional groups, denoted by the colours: red (phytoplankton), yellow (detritus), cyan (DOM:Sem-C, DIC and TempC), blue (SiO4, NO3, PO4), pink (HTL and DOM) and grey (Salinity, Mesozoo-C, DOM:Ref-C, NH4, Bio-Alk).



# 5   Conclusions

Marine biogeochemistry is complex to simulate, representing a plethora of processes, in an often computationally costly man-
ner. As a result, it is not well-suited for addressing specific questions that necessitate extensive and long-lasting ensemble
simulations such as ecosystem response to climate change and anthropogenic stresses across a broad range of scenarios, or
analyses aimed at informing policy-making decisions. Here, we aimed to use complex network analysis to gain insight into
connections found across the ecosystem, while providing understanding that will aid in the simplification of its complex inter-
actions and dynamics.

With future observation missions that will provide new biogeochemical variables for assimilation, there is a need to further
understanding of how transferable the correlations lengthscales are between the different biogeochemical variables. Using the
correlation analysis and the resulting spatial networks, we can conclude that the biogeochemical lengthscales vary significantly
between variables and are not directly transferable. However, we have provided an approximation for the lengthscale of each
variable, and each spatial location, that is informed by the high correlation in the spatial variability between lengthscales of each
variable. With this clear indication of structure embedded into the horizontal connectivity of the ecosystem, we sought to split
the shelf sea into geographic regions using clustering network algorithms. This clustering process was applied to each variable
independently, yet it identified a set of clear and consistent boundaries that represent areas of extremely low connectivity
across which information is not shared. This resulted in 13 key regions, suggesting that each functions as a quasi-separate
system, but with unified biogeochemical/ecosystem characteristics within its boundaries. This also identified the Celtic Sea
and the north-west section of the NWES as areas of high shelf-sea to open ocean exchange. Finally, we demonstrated that
the complex network carries important information on how the ecosystem variables cluster into natural "functional types".
Our analysis demonstrated that the chemical components (e.g., nitrogen, carbon, silicon...etc) of each pelagic variable (e.g.,
diatoms, nanophytoplankton, microzooplankton) are closely linked and a simpler version of the model can be built, by reducing
these variables through parametrization. We also see that the pelagic variables form even larger functional groups (e.g., POM,
phytoplankton, HTL/DOM), composed of variables that can be effectively parametrized through monotonic functions of each
other.

These findings show that complex networks can be used as an effective tool in dismantling the complexity of the ecosystem
dynamics, providing simplifications to the system extracted from the behaviour of the model itself. These simplifications will
be applied in future work, e.g., aimed at building an ML-based reduced order emulator to improve data assimilation on the
NWES.

*Data availability.* Data is available on MASS and obtainable on request. MASS is the Met Office Managed Archive Storage System and is
accessed using the user interface known as MOOSE, access is now possible from both MONSooN and the JASMIN system at CEDA.





*Author contributions.* IH written and executed all code. JS provided model and data. All contributed to analyse and interpret the results, proof-reading and adjusting the text.

*Competing interests.* No competing interests are present. SC is a member of the editorial board of journal "Biogeosciences". The peer-review process was guided by an independent editor, and the authors have also no other competing interests to declare.

*Acknowledgements.* IH acknowledges the support of the Natural Environment Research Council via the National Centre for Earth Observation (Contract Number PR140015), and the University of Reading (NCEO/Reading 26). JS and SC were supported by the projects SEAMLESS (funded by the European Union's Horizon 2020 research and innovation programme under grant agreement No 101004032)
and NECCTON (funded by the Horizon Europe research and innovation action under grant agreement No 101081273). AC acknowledges the support of the project SASIP (grant no. 353) funded by Schmidt Futures – a philanthropic initiative that seeks to improve societal outcomes through the development of emerging science and technologies. RB acknowledges the support of the NCEO (contract number PR140015).



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
