# Peer review of "Investigating ecosystem connections in the shelf sea environment using complex networks"

_EGUsphere, 2023_

## Referee Comment (RC2)

**Overview**

In this paper, the authors use complex network theory with outputs from a model simulation of the North-West European Shelf (NWES) to identify 1) spatial correlation length scales of biogeochemical variables, 2) geographical regions with strong spatial correlation within them and weak correlation between them and 3) correlations between biogeochemical variables. Point 1) is achieved by computing the Spearman's correlation coefficient between the time series of the different grid points. For point 2), for each variable, they build a spatial network with the previous coefficient, apply spectral graph clustering to gather grid-points and identify the boundaries of thieseclusters. Then, they define the regions base on the fraction of variables that have a boundary in each grid point. For point 3), they compute the Spearman's correlation coefficient between the spatial distributions of each variable, build a spatial network with that and use the spectral graph clustering to cluster biogeochemical variables. A first result of this work is to show that complex network theory can be used to identify biogeochemical regions based on spatial correlation or to identify correlation between biogeochemical variables. This is of interest for reducing the complexity of biogeochemical dynamics and for helping the analysis of simulations. The correlation length scales are of interest for data assimilation as it quantify the range of the influence between grid points.

I very much appreciated to read the paper. It is clear and well written. The results are of interest and worth to be published. It presents an interesting way to analyse biogeochemical model outputs. The definition of biogeochemical provinces is particularly interesting as it can help the analysis of models. The methods are clearly explained. I do not have major comments on the paper, but rather a list of minor or specific comments that I think could further improve the paper. The comments that are more important are highlighted in red.

As a summary of my comments, here are my answers to the review criteria at Biogeosciences. I just selected the relevant questions:

1. Do the authors give proper credit to related work and clearly indicate their own new/original contribution? *Yes. Maybe a bit of comparison with the literature on correlation length scales could benefit the paper.*

2. Does the abstract provide a concise and complete summary? *Mostly. It could be improved by more clearly stating the results*

**Minor and specific comments**

Abstract

I think the results should be more clearly/precisely stated in the abstract. It seemed a bit to vague to me. For example:

- l. 4: « to identify the functional types », which one are they exactly?

- l. 6: « identifying the (geographically varying) connectivity lengthscales and the clusters of spatial locations that are connected. » What are the main findings concerning the length scales? What are the different clusters? For the length scales, results that seems particularly interesting is that spatial variability is quite similar between variables, requiring only to scale it using the mean length.

- l. 9: « The results of this study help to understand how natural, or antrophogenic, perturbations propagate through the shelf-sea ecosystem », it is difficult to agree with that last sentence since the results where not clearly stated before. After finishing reading paper, I also do not think the results help to understand how perturbations propagate in the ecosystem. The results rather offer a analysis framework to do that.

- l. 9: « antrophogenic » -> anthropogenic

**Introduction**

l. 35: « an abstraction that will allow for smarter decision-making when considering data sampling and feature selection for ML. » Not that clear to me how and why abstraction can allow smarter decision-making.

l. 37-50: Very nice paragraph clearly stating the objective of the work.

**Model and Data**

Sec. 2.1: I think it will be nice to have a bit more details about the configuration. Things like: numerical schemes, diffusion, viscosity, equation of state, what forcings (wind, temperature?). How the simulations are run (spin-up procedure, initialisation…). The reference to the papers should be for further details. The reader should not need to read these papers to get a basic understanding of the configuration.

**Methodology**

Sec. 3.1: maybe a figure showing the raw and filtered time series in the supplementaries could be useful to illustrate what are the timescale filtered? Or maybe some periodogram? It should probably be stated before (introduction? Or somewhere in the methods?) what are the timescales of interest? And why? Out of curiosity have you tried your analysis with the seasonal signal?

l. 154: « to a 21 km spatial resolution » make me wonder if the results are sensitive to the resolution of the model? Longer length scale because of eddy mixing? Or shorter one because of dynamical barrier created by filaments or eddies? This somehow questions also the isotropy assumption.

l. 162: I do not understand why the authors say : « As opposed to the biogeochemical lengthscales computed in Sect. 3.2.1 […] here we manipulate the spatial networks to look at the spatial dependency of this length scale. » In section 3.2.1 you also have a map of the length scales that give you the spatial information (Fig. 2). I do not get the interest of these two definitions. Note that this also bring a bit of confusion about which are the length scales used for the different plots. For example in Fig. 4 which one is it? And for Fig. 5? I kind of got that Fig. 4 is  the length scale define in sec. 3.2.1 and Fig. 5 the one in sec. 3.2.3 but it is not so clear.

l. 167: « black » rather than « red »?

Sec. 3.3: This part is not easy to follow. Maybe a short description of the objective at the beginning could help the reader. What are the objects to be clustered, following which criteria? If I understood well, the goal is to clusters grid-points depending on their temporal correlation between each other for each variables so that grid-points with strong correlation are group together.

**Results and Discussion**

Sec. 4.1: As mentioned before, mentioning which length scale (the one from sec. 3.2.1 or sec. 3.2.3) the authors refer to would help the reader. Since two definition of length scale seems to be used, it feels natural to wonder how they compare?

l. 275-278: I think I got the general idea here: the spatial distribution of the length scale of a specific variable is the product between Fig. 5a and Fig. 4. However, as it seems that it is not the same definition of the length scale between Fig. 4  and Fig. 5a it is a bit confusing.

Sec. 4.1: I am not familiar with length scale, but it seems that there is some literature on length scales (just saying that based on a quick search on google scholar). Some comparison of the results and the methods with the literature is missing there. Are there other definition of length scale? How does the method used in this paper compare with other? Are the length scales similar to former estimations?

Fig. 7: How is it done? I guess it is some kind of generalisation of Fig. 6 but it would be good to know more than « We used those robust boundaries to identify 13 regions representing areas of NWES connectivity. Results of this regionalisation are represented in Fig. 7. » (line 315)

l. 350: « or build simpler models than ERSEM » I think this need to be say a bit differently. Complexity of models tends to increase to  better (or hoping to better) represent the real world. NPZD models already exist with just one phytoplankton, one zooplankton… Here the issue is to simplify ERSEM while keeping an accurate representation. Maybe something like line 51 « simplified (yet realistic with respect to the objectives) ».

l. 363-366: I do not see that in Fig. 8. The mean correlation between POM (yellow) and the Higher Trophic Levels + DOM (pink) is rather low. The authors should clarify.

Conclusions

l. 410-426: You are here a bit more specific about the results and this could be used for the abstract. E.g. « we can conclude that the biogeochemical lengthscales vary significantly between variables and are not directly transferable. » or « we have provided an approximation for the lengthscale of each variable, and each spatial location, that is informed by the high correlation in the spatial variability between lengthscales of each variable »…

l. 421-424: « Our analysis demonstrated that the chemical components (e.g., nitrogen, carbon, silicon. . . etc) of each pelagic variable (e.g., diatoms, nanophytoplankton, microzooplankton) are closely linked and a simpler version of the model can be built, by reducing these variables through parametrization. » I do not know ERSEM but I assume that as many models it started from a simple version and the complexity has been increased (e.g. addition of more phytoplankton types). I am wondering how the grouping compare with a former simpler version of ERSEM? I suppose it should be relatively similar (e.g. all types of phytoplankton in gather in only one) however it will be quite interesting if some grouping where different.

Extra comments

« lengthscales »: After a quick search on google scholar, it seems that it is rather written « length scales » or « length-scales ».

The regions define in Fig.  7 could be used for sampling the domain to analyse the inter-variable interaction network. Maybe selecting grid points only within one region and to compare with the same done with another region. Are the interaction between variables different between two regions? Or sampling evenly between the regions to have a fair general representation? This point is mostly for curiosity as it seems natural to try to use these regions.

l. 367: Butenschon et al. (2015) and Butenschon et al. (2016) are similar paper (2015 is the discussion version of 2016). Better to keep only 2016.

---

## Author Comment (AC1)

Thank you for taking the time and care to provide valuable feedback and contributions to this manuscript. Please see our responses to the comments below, which we are ready to implement for a future revision.

Copy of review comments (RC) are given below, followed by the author comments (AC).

**Responses to RC1:**

| RC00 | This paper analyses output from a complex biogeochemical model, ERSEM, using network analysis.  The analysis is used for several purposes: evaluating the spatial length scale of the variables, determining areas of coherent biogeochemical interactions and boundaries of low connectivity, and establishing which variables are highly connected with each other. This information is useful when setting up regional systems. and evaluating the interactions between model variables and weather the system can be approximated well by a simpler representation.   The length scales are useful in data assimilation systems, when setting the area of influence of the observations. I think the paper provide new knowledge worth publishing, but before I would like the following points addressed: |
|---|---|
| RC01 | - Only surface data is used, this is reasonable to reduce the amount of data, but it would require a discussion of the implications of such a choice.  For example, in the resulting network from the analysis (Figure 9) the detritus is completely disconnected from the photo and zooplankton, but as that quickly sinks out it would not remain one on the surface and maybe using only surface data is the reason for this disconnect? There is also a question wether there are other methods to reduce the data size that would retain more information throughout the water-column that could have been used? |
| AC | We thank the reviewer for pointing this issue out and we will write a paragraph transparently discussing this limitation immediately after introducing the dataset we used throughout the manuscript (currently section 2, line 75). It should be noted that we expect the method of reducing the   data to the surface, will only impact the variable cross-correlation matrix in Fig. 9.
We believe that choosing the surface data is useful, as:
(i) it is directly relevant to DA horizontal lengthscales near the surface, which is the most observed part of the water column on the NWES.
(ii) it transparently captures ecosystem connections in the mixed layer, which is the most biologically active part of the ocean.
(iii) in this mixed layer, we capture the lengthscales for probabilistic modelling (ensemble development).
We do not know a better methodology that could retain advantages of points (i), (ii) and (iii) in a computationally affordable way. |
| | |
| RC02 | - The longer time-scales are filtered out, so there could be biogeochemical feedback mechanisms that work on timescales >10 days that are filtered out. So what happens when resulting network is used to inform an emulator, and then applied in the context of climate as suggested by the authors?  This also needs to be addressed in the discussion. |
| AC | While we could imagine a strong connection within sub-monthly timescales that gradually weakens (or disappears) on long (e.g. climate) timescales, in this case it is hard for us to think of a physical or biogeochemical process that would cause this, making it quite unlikely.
We filter the seasonal timescales as they superficially increase the correlation thresholds, due to the seasonal harmonics common to many variables.  We can think of this filter as a sort of "normalization" technique that makes the results more human understandable, rather than something that affects the analysis of connectivity (it will not affect which |

| | |
|---|---|
| | variables are more correlated and which less, but it will lower the overall correlation between all the variables).
 We cannot check what happens on climate timescales, but we argue that the connections captured on the short-time lengthscales (e.g. the differences between more and less connected variables and regions) of the high-pass filtered data are very likely the dominant factor in any (non-filtered) long-term connectivity analysis.
 While we fully agree that caution is required for any machine learning model trained on local data to be applicable on a climate scale, we would expect the simplifications and relationships learned from the complex network (this can be applied to e.g. select types of features for the climate emulator) to survive.
 To summarise, we believe that these connections could be used to design a climate emulator, but the emulator would need to be ideally trained on climate data. This is a long-term goal of some ongoing research (e.g. at PML), which is why we mention it in the manuscript. |
| | |
| RC03 | - Applicability of results: Would this results of the analysis be valid other models? For example could the length scales obtained be used in data assimilation system using another BGC model than ERSEM? Would the length scales apply when assimilation observations deeper in the water column even if your results that are only based on surface model data? |
| AC | The methodology highlighted by this manuscript is naturally applicable to different models and datasets. We would expect the key characteristics derived here from ERSEM to be representative of ecosystem itself, and therefore of other trustworthy representations of the ecosystem (i.e. trustworthy models). Wherever the derived characteristics could be compared to the general knowledge, they compared very well. We note that there are severe constrains on validating our results with observations, due to the intermittency of satellite observations - and doing the same analysis with other models is beyond the scope of this project. This has been discussed in the manuscript, please see the line 70.
 We would expect the lengthscales to be broadly applicable in the mixed layer of the ocean, and to be not applicable beneath the mixed layer. The majority of biological growth happens in the mixed layer, and so it is of higher significance and relevance to investigate (we might think differently if we were instead investigating global or deep ocean systems). In addition to this, on the NWES there aren't that many sub-mixed-layer observations, at least compared to the number of surface observations, making our knowledge of horizontal lengthscales at these depths more challenging, and not our particular focus. |
| | |
| RC04 | - The description of the methods could be improved for the benefit of the reader, I provide some suggestions for what needs to be clarified below. |
| AC | The suggestions are extremely helpful and go to strengthen the manuscript. Details of each improvement are given with the corresponding suggestion. |
| | |

| | Specific comments |
|---|---|
| RC05 | Title: Could the title be improved but adding "Investigating" at the beginning? |
| *AC* | Yes agreed. We will change the title accordingly. |
| | |
| RC06 | Abstract:
The expression "functional types of variables" is used in the abstract and in the text, it is a bit unclear to me what this means. The expression becomes particularly confusing since the ERSEM itself also includes functional types of plankton. Consider either using a different expression or define it properly before using it. |
| AC | Different expression to be used: "functional groups" when referring to a group of similarly behaving variables, to avoid confusion with PFTs used by ERSEM. |
| | |
| RC07 | "Be also used" should be "also be used" |
| *AC* | Thanks for noticing this, we will change as suggested. |
| | |
| RC08 | What is meant by "flow of information between degree of freedom" |
| AC | Will rewrite to say:
"within which there is a large exchange of information within the ecosystem" |
| | |
| RC09 | The first part of the last sentence is unclear to me: I don't see that it is demonstrated anywhere how these results can be used to understand how a perturbation propagate through the ecosystem. |
| AC | Will rephrase :
"The results of this study describe how information is expected to propagate through the shelf-sea ecosystem on the time-scale of interest, and how it can be used in multiple future applications such as stochastic noise modelling, data assimilation, or machine learning."

Some examples include:
The horizontal lengthscales help to identify areas where the information is simultaneously shared across space.
The regionalisation indicate that an area has some level of shared behaviour, meaning information from these regions are more likely to have a stronger influence within the region than outside of the regions.
The inter-variable analysis indicates how information of a particular variable can spread to other linked/clustered variables, as certain subsets of the state variables are shown to behave with a strong correlation. |
| | |
| RC10 | Line 38: "…investigate three relevant questions related …" either formulate the three topic as questions or rewrite the sentence on line 38. |
| AC | We will rewrite the sentence on line 38 to read "…investigate three relevant *topics* related …" |
| | |
| RC11 | Line 40: "based on" should be "apply". |
| AC | Thanks, and agreed. We will change accordingly. |
| | |
| RC12 | Line 40: Is this length scale only useful when applying variational data assimilation, not other (ensemble) data assimilation techniques? |
| | Thanks, this is correct. The results are also highly relevant to ensemble data assimilation (EnDA) too. For instance, it can guide scale-aware localisation for ensemble-based error covariance. We will add text on line 40 to highlight this applicability. |
| | |

| | |
|---|---|
| RC13 | Line 49: as mentioned before, the use of the expression the use of the expression "functional type" is a bit confusing, please define it here. |
| AC | We will rephrase this expression in line with our earlier comment on "functional types" ->" functional groups". Here we will also define a functional group (for additional clarity): "functional groups (i.e. a set of state variables that are generally highly correlated with each other)." |
| | |
| RC14 | Line 51: The statement that these traditional biogeochemical models are unsuitable to address response to climate change, effectively writing off all CMIP simulations is quite severe, I would suggest to moderate the statement. However I do agree that lighter model systems are more suitable for ensemble simulations, but it they are trained on data from the present day, they may not be very good at representing future ecosystem response. |
| AC | As well as line 51, we believe the reviewer may also be referring to line 21, which states: "such as ecosystem's response to climate change and anthropogenic pressures across large variety of scenarios".
 We agree and recognise that these large models can display a high degree of realism, but the computational burden of these simulations means there is also a need for reduced order models that can cover a "large variety of scenarios" (where these ESMs may only explore some subset of this range).
 The current text may have led to this misunderstanding, so we will tidy up the language to avoid this. |
| | |
| RC15 | Line 88: Were the river nutrients also included and were they also annual? |
| AC | Yes, the rivers included nutrients, varying daily. We will provide this information in the manuscript. |
| | |
| RC16 | Line 120: the transformation to the time-local standardised form is very well explained, but I wonder what happens in period when standar deviation is low or zero (for example I winter), does and stay finite? |
| AC | Thanks. The aim is to create a data set with a uniform standard deviation (i.e. unity). When sigma is small (denominator in (3)), the numerator is small also. The values remain finite. |
| | |
| RC17 | Line 120: Would river input influence the network results, for example would there be a stronger connection between the biogeochemistry and salinity in a region of strong river influence. I.e. would the network presented in figure 9 differ from region from region to region? |
| AC | The river inputs influence the network structure, which is clear from Fig. 7 "Region G" – a region largely defined by the delta of River Elbe (see also the difference in nutrients in supplementary material).
 In the context of Fig. 8, we would expect certain connections to be amplified/dampened if we were to reduce the sample size to a specific region. The figures provided aim to show an overview for the system behaviour, acknowledging these effects with the coefficients of variation shown in Fig.8. While we acknowledge that there is a great variety of additional questions one can ask and explore, in this work, we don't have scope to address all of these questions and we have to be selective (e.g. we don't specifically address the difference between river delta areas and areas further from the coast). |
| | |

| | |
|---|---|
| RC18 | Line 124: I did not see it specified anywhere that data were treated any differently, so could you just simply write that all dat were treated this way? |
| AC | Agreed. Will rewrite to simply state:
 "All data used in this study have been preprocessed using the procedure in Eqs…" |
| | |
| RC19 | Sections 3.2.1: Biogeochemical length scale estimation: What did you do in regions close to land or the boundary? Did you not compute the length scale or only consider the ocean points?  The same question applies to the method in 3.2.2 |
| AC | In Sec. 3.2.1, we only considered the ocean points away from the boundary for this initial calculation of the average lengthscale of each variable. We will make this clear in the manuscript.
 In section 3.2.2, we upscale from a 7km -> 21km grid. We use the arithmetic mean of the relevant points to achieve this upscaling. To account for the boundaries, we consider a 21km grid point to be ocean only if more than half of the relevant 7km points are also ocean. In addition, this provided good results, and the lengthscales at the boundaries are discussed in section 4.1:
 "It is notable that another area of low-connectivity is the open (Atlantic) AMM7 domain boundary regions. This indicates that the boundary conditions of the regional model de-correlate from the rest of the domain…" |
| | |
| RC20 | Difference between method in 3.2.1 and 3.2.2: Am I correct that the difference between 3.2.1 and 3.2.2 is that 3.2.1 is done on a finer grid and uses a different method to compute the length scale?  The coarsening before computing the length scale is primarily used to reduce the amount of data given to the SGC?  Is this correct or are there other resort to compute length-scales twice?  This could be made clear in the manuscript. |
| AC | Thank you. Your conclusion is mostly correct, and we shall make this clearer in the manuscript. Some slight clarifications:
 Section 3.2.1 is used to determine the average lengthscale of each variable.
 Section 3.2.2 is used to determine the average lengthscale of each geographical point (independent of the specific variable). Since Sec. 3.2.1 shows that the average lengthscales are different, we account for this by using the different correlation coefficient thresholds on each variables network, such that each network has the same number of links. This effectively normalises the lengthscales for each variable, and allows us to calculate an average for each point.
 While we allude to this in the opening sentence of section 3.2.3:
 "As opposed to the biogeochemical lengthscales computed in Sect. 3.2.1, which refer to each variable and reflect their physical properties averaged on the domain, here we manipulate the spatial networks to look at the spatial dependency of this length-scale."
 We will make a statement earlier in the manuscript to make this clearer. |
| | |
| RC21 | Line 153: How was the grid upscaled from 7 to 21 km? |
| AC | Will add text to specify that "arithmetic averaging" was used. |
| | |
| RC22 | Line 154-160 Explanation of pruning: This is very hard to understand, please explain better how this was done. |
| AC | We will revise text used to explain the pruning will be revised to make it clearer.
 We will likely re-phrase the relevant text, and move it to a more appropriate section of the manuscript (Sec. 3.2.3) instead of in the section for initially generating the spatial networks (3.2.2). |
| | |

| | |
|---|---|
| RC23 | LIne 170 from "We took …" and the next sentence mean exactly the same thing. Remove the first sentence (or last, up the author, but I preferred the last). |
| AC | We will remove the first sentence (and keep the last). |
| | |
| RC24 | Before line 180: This is not easy to understand, could you please try to make this clearer: "This was done by taking the mean lengthscale at each grid point across all variables from the dynamically thresholded spatial networks. In order to assess whether this spatial variation could be well approximated by the mean of these lengthscales, we compared the spatial distribution of lengthscales between each different variable using Pearson's correlation. Here, we would expect to see a high correlation if the structure of the spatially varying lengthscales is consistent. This set of spatially varying lengthscales was then represented as a ratio of the mean." |
| AC | Agreed that this is not clear. We will rewrite this paragraph to make it much clearer. |
| | |
| RC25 | Line 193: "a links … defined by the Spearman correlation.. " at this point there has been introduces severe spearman correlation, the length scale of the correlation with itself on a 7 km grid, the length-scale om a 21 km grid and the correlations between the length scales of different variables, so which one does this refer to here? |
| AC | We will adjust the text to make specific reference to the relevant case: "defined in this case by the Spearman's correlation between each node on the 21km grid )" |
| | |
| RC26 | Paragraph line 190-200: Please write out the equations on its own line (as on page 7) and give them numbers to benefit the reader. |
| AC | Agreed, we will give the equations their own lines to make it clearer for readers. |
| | |
| RC27 | Line 220: This is difficult to follow: "In order to compare the regionalisation of each variable, we first projected the cluster labels of each node back onto the horizontal plane. Then, we applied an edge detection kernel to identify the boundaries between differently labelled regions, creating a boundary map for each variable (with value 1 at boundary grid points and 0 elsewhere)." Please refer back to the appropriate equation on the previous page (ref. my comment above). |
| AC | We will rewrite this to make it clearer, and refer to the relevant clustering equation: We identified ``robust regions'' as connected areas of ocean that rarely, or never, contain the boundaries from the clustering of any individual state variable. For the spatial network of each variable, we identified every node that is geographically adjacent to another node with a different cluster label (as found from Eq. (X)). These nodes represent the boundaries between different regions. Since each node in a spatial network will have a corresponding node in the spatial network of every other variable (i.e., they share the same geographical point), we could then calculate the frequency with which each geographical point occupies a boundary node, across all ERSEM state variables. These ``boundary frequency'' values are then plotted onto a grid, according to their geographic location, so that the robust regions can be identified visually. |
| | |

| RC28 | Line 235: You calculate the mean adjacency matrix over 300 point randomly selected over the shelf <200 meter and then average that. Then later you say "the boundaries particularly seem to reflect shallower bathymetry (approx. 100 m) than the 200 m depth usually applied to delimit the margins of shelf-seas, including NWES." So why not samle within 100 meters? |
|---|---|
| AC | We chose to use the 200m bathymetry as it is the standard convention to define shelf-seas, including the NWES (Skakala et al (2022), Huthnance et al (2009), Borges et al (2006). This delimitation corresponds to the continental shelf. The 100m area highlights that a big part of this region is linked to the open North Atlantic, which is of interest.

*Skákala, J., Bruggeman, J., Ford, D., Wakelin, S., Akpınar, A., Hull, T., Kaiser, J., Loveday, B.R., O'Dea, E., Williams, C.A. and Ciavatta, S., 2022. The impact of ocean biogeochemistry on physics and its consequences for modelling shelf seas. Ocean Modelling, 172, p.101976.*

*Huthnance, J.M., Holt, J.T. and Wakelin, S.L., 2009. Deep ocean exchange with west-European shelf seas. Ocean Science, 5(4), pp.621-634.*

*Borges, A.V., Schiettecatte, L.S., Abril, G., Delille, B. and Gazeau, F., 2006. Carbon dioxide in European coastal waters. Estuarine, Coastal and Shelf Science, 70(3), pp.375-387.* |
| | |
| RC29 | Line 255: Be precise: inclusion of new types of observations *for assimilation* … |
| *AC* | Agreed. This change will be implemented. |
| | |
| RC30 | Line 255: I suggest to remove "profound". |
| *AC* | Agreed. This change will be implemented. |
| | |
| RC31 | Line 265: suggest: "oxygen have different lengthscales …" |
| *AC* | Agreed. This change will be implemented. |
| | |
| RC32 | Line 379: "… we applied SGC…": did you also test different values of k here? |
| *AC* | Yes, we mention this on lines 217. |
| | |
| RC33 | Line 390: "Ammonium dynamics are relatively more complex than the ones of nitrate." This sentence can be removed. |
| AC | Agreed. This will be done. |
| | |
| RC34 | Figure 9: How was the lines connecting the different variables decided? |
| AC | Added text to Figure 9 caption:
"The highest correlations (top 25\%) of all possible pairwise correlations between variables are shown (grey lines)." |
| | |
| RC35 | Line 427: I suggest to use another word than "dismantling". |
| AC | We will change to "simplifying" instead. |
| | |
| RC36 | Concerning the supporting information, this would be easier to understand if the variables plotted were given standard names and the y-axis were supplied with the units. |
| AC | We will change the plots in the supporting information, re-format them with standard names, and correct units on the y-axis. |
| | |

Let us thank again the reviewer for their important suggestions and we hope that after the suggested changes, addressing the reviewer comments, the manuscript will be in a good shape to be accepted for publication.

Best wishes,

Ieuan Higgs and the co-authors

---

## Author Comment (AC2)

Thank you for taking the time and care to provide valuable feedback and contributions to this manuscript. Please see our responses to the comments below, which we are ready to implement for a future revision.

Copy of review comments (RC) are given below, followed by the author comments (AC).

**Reviewer 2:**

| | |
|---|---|
| RC40 | Overview
In this paper, the authors use complex network theory with outputs from a model simulation of the North-West European Shelf (NWES) to identify 1) spatial correlation length scales of biogeochemical variables, 2) geographical regions with strong spatial correlation within them and weak correlation between them and 3) correlations between biogeochemical variables. Point 1) is achieved by computing the Spearman's correlation coefficient between the time series of the different grid points. For point 2), for each variable, they build a spatial network with the previous coefficient, apply spectral graph clustering to gather grid-points and identify the boundaries of these clusters. Then, they define the regions base on the fraction of variables that have a boundary in each grid point. For point 3), they compute the Spearman's correlation coefficient between the spatial distributions of each variable, build a spatial network with that and use the spectral graph clustering to cluster biogeochemical variables. A first result of this work is to show that complex network theory can be used to identify biogeochemical regions based on spatial correlation or to identify correlation between biogeochemical variables. This is of interest for reducing the complexity of biogeochemical dynamics and for helping the analysis of simulations. The correlation length scales are of interest for data assimilation as it quantify the range of the influence between grid points.
I very much appreciated to read the paper. It is clear and well written. The results are of interest and worth to be published. It presents an interesting way to analyse biogeochemical model outputs. The definition of biogeochemical provinces is particularly interesting as it can help the analysis of models. The methods are clearly explained. I do not have major comments on the paper, but rather a list of minor or specific comments that I think could further improve the paper. The comments that are more important are highlighted in red (see the pdf file attached for colored version). |
| | As a summary of my comments, here are my answers to the review criteria at Biogeosciences. I just selected the relevant questions: |
| RC41 | 1. Do the authors give proper credit to related work and clearly indicate their own new/original contribution? Yes. Maybe a bit of comparison with the literature on correlation length scales could benefit the paper. |
| AC | We agree. We will include some comparison to addition literature on lengthscales. Some examples are given below:
*Fowler, A.M., Skákala, J. and Ford, D., 2023. Validating and improving the uncertainty assumptions for the assimilation of ocean-colour-derived chlorophyll a into a marine biogeochemistry model of the Northwest European Shelf Seas. Quarterly Journal of the Royal Meteorological Society, 149(750), pp.300-324.*

*Desroziers, G., Berre, L., Chapnik, B. and Poli, P., 2005. Diagnosis of observation, background and analysis-error statistics in observation space. Quarterly Journal of the Royal Meteorological Society: A journal of the atmospheric sciences, applied meteorology and physical oceanography, 131(613), pp.3385-3396.*

*Hollingsworth, A. and Lönnberg, P., 1986. The statistical structure of short-range forecast errors as determined from radiosonde data. Part I: The wind field. Tellus A, 38(2), pp.111-136.* |
| | |

| | |
|---|---|
|  | 2. Does the abstract provide a concise and complete summary? Mostly. It could be improved by more clearly stating the results |
| AC | Thanks. The abstract will be amended according to the "minor and specific comments" to address this. |
| | |

| | |
|---|---|
| | Minor and specific comments |
| | Abstract |
| | I think the results should be more clearly/precisely stated in the abstract. It seemed a bit to vague to me. For example: |
| RC43 | - l. 4: « to identify the functional types », which one are they exactly? |
| AC | Different expression to be used: "functional groups" when referring to a group of similarly behaving variables, to avoid confusion with PFTs used by ERSEM.
We will also mention the key groups in brackets after (i.e. phytoplankton, detritus and heterotrophs & DOM). |
| | |
| RC44 | - l. 6: « identifying the (geographically varying) connectivity lengthscales and the clusters of spatial locations that are connected. » What are the main findings concerning the length scales? What are the different clusters? For the length scales, results that seems particularly interesting is that spatial variability is quite similar between variables, requiring only to scale it using the mean length. |
| AC | We will update the abstract to give more detail on the lengthscale results.
"We show that the spatial correlation lengthscales vary significantly between variables and are not directly transferable, however they are distinguished only by a constant scaling factor: the spatial distribution of lengthscales is similar for each variable."
We may update the abstract to include some of the specific regions (e.g defined by river input, or open-ocean to shelf-sea exchange), although we identify 13 different regions/clusters in our analysis, each only labelled with a letter A-M. Detailing all of them would seem unnecessarily specific in the case of the abstract. |
| | |
| RC45 | - l. 9: « The results of this study help to understand how natural, or antrophogenic, perturbations propagate through the shelf-sea ecosystem », it is difficult to agree with that last sentence since the results where not clearly stated before. After finishing reading paper, I also do not think the results help to understand how perturbations propagate in the ecosystem. The results rather offer a analysis framework to do that. |
| AC | We agree that our method provides a framework to describe the propagation of information. We also agree that using the word "understand" in relation to how perturbations propagate was perhaps too strong and we will change this to "describe". The use of "describe" can be justified since:
• The horizontal lengthscales help to identify areas where the information is shared across space.
• The regionalisation indicate that an area has some level of shared behaviour, meaning information from these regions are more likely to have a stronger influence within the region than outside of the regions.
• The inter-variable analysis indicates how information of a particular variable can spread to other linked/clustered variables, as certain subsets of the state variables are shown to behave with a strong correlation.
The existing statement will be rephrased as: |

| | |
|---|---|
| | "The results of this study describe how information is expected to propagate through the shelf-sea ecosystem on the time-scale of interest, and how it can be used in multiple future applications such as stochastic noise modelling, data assimilation, or machine learning." |
| | |
| RC46 | - l. 9: « antrophogenic » -> anthropogenic |
| AC | *Agreed, thanks for pointing this out.* |
| | |
| | Introduction |
| RC47 | l. 35: « an abstraction that will allow for smarter decision-making when considering data sampling and feature selection for ML. » Not that clear to me how and why abstraction can allow smarter decision-making. |
| AC | We agree that the word abstraction" is inappropriate and misleading here, we will replace it with the word "information". The key message of this sentence is that identifying connections across the NWES and variables, would indicate which variables and locations are unneeded as input features into ML algorithms. |
| | |
| RC48 | l. 37-50: Very nice paragraph clearly stating the objective of the work. |
| AC | *Thanks.* |
| | |
| | Model and Data |
| RC49 | Sec. 2.1: I think it will be nice to have a bit more details about the configuration. Things like: numerical schemes, diffusion, viscosity, equation of state, what forcings (wind, temperature?). How the simulations are run (spin-up procedure, initialisation...). The reference to the papers should be for further details. The reader should not need to read these papers to get a basic understanding of the configuration. |
| AC | We agree, and we will give more details about the configuration used. |
| | |
| | |
| | Methodology |
| RC50 | Sec. 3.1: maybe a figure showing the raw and filtered time series in the supplementaries could be useful to illustrate what are the timescale filtered? Or maybe some periodogram? It should probably be stated before (introduction? Or somewhere in the methods?) what are the timescales of interest? And why? Out of curiosity have you tried your analysis with the seasonal signal? |
| AC | We agree and we will provide a Figure comparing the raw and filtered time-series in the Supplementary Information. The time-scales are limited by the resolution of outputs (1 day) on one end, and by the need to remove seasonality on the other end, since seasonality introduces artificial correlations. The links between variables are explored through multi-year simulations, since these we could computationally afford. We however believe that the links between variables and regions that we identified here could be applicable to longer time-scales than the time-scale of the simulation. Note, some tests were done using the seasonal signal, however as already said this signal results in a very large temporal correlation across the entire domain, obfuscating any detail within the region. |
| | |
| RC51 | l. 154: « to a 21 km spatial resolution » make me wonder if the results are sensitive to the resolution of the model? Longer length scale because of eddy mixing? Or shorter one |

| | |
|---|---|
| |  |
| AC | As for the nature of the horizontal lengthscales identified in the manuscript: We expect that physics (eddy-mixing, filaments, eddies, which should be resolved by the 7km model) is important contribution to the length-connections (e.g. increase of Rossby radius in the open ocean), but one needs to keep in mind that for the biogeochemical variables these drivers are intertwined with biogeochemical drivers that will also be reflected in the output (e.g. river delta geography, interaction of sunlight with biology…etc).

To clarify the 7km and 21km resolutions: the model is run at 7km resolution, and only the daily model outputs are upscaled to 21km, because the 21km is the highest resolution we could feasibly use in our complex network analysis for computational reasons. When it comes to the model resolution (7km), obviously using higher resolution than 7 km would improve the physics of the model (e.g. increase of the model resolution to 1.5km is an ongoing stream of work at multiple involved institutions), but for the spatial scales considered by this manuscript (I.e. the NWES-wide analysis)  it is widely accepted that 7 km model resolution provides a good approximation to physics and biology. At the end, the 7 km model is used operationally at the UK Met Office and its outputs are supplied to CMEMS. To summarize, we expect that increasing spatial resolution would lead to only higher order corrections to the results presented in this work. |
| | |
| RC52 | l. 162: I do not understand why the authors say : « As opposed to the biogeochemical lengthscales computed in Sect. 3.2.1 [...] here we manipulate the spatial networks to look at the spatial dependency of this length scale. » In section 3.2.1 you also have a map of the length scales that give you the spatial information (Fig. 2). I do not get the interest of these two definitions. Note that this also bring a bit of confusion about which are the length scales used for the different plots. For example in Fig. 4 which one is it? And for Fig. 5? I kind of got that Fig. 4 is the length scale define in sec. 3.2.1 and Fig. 5 the one in sec. 3.2.3 but it is not so clear. |
| AC | It is correct that 3.2.1 relates to Fig. 4, and Sec. 3.2.3 relates to Fig.5.
We will add some text to Fig. 2's captions to make clearer – this is just a visualisation of what a length scale calculation looks like.
The difference here, is that 3.2.1 calculates the average length-scale of each variable, while the networks used in section 3.2.3 aim to look at the way that these lengthscales vary spatially. Utilising the network structure (with pruning and totalling the number of connections) allows us to effectively normalise these lengthscales, so we can directly compare the spatial distribution of each variable to each other. We will make these need for these differences clearer in the text. |
| | |
| RC53 | l. 167: « black » rather than « red »? |
| AC | Agreed. Will rephrase to:

"shows a set of nodes (red) connected to the current target node (black)" |
| | |
| RC54 | Sec. 3.3: This part is not easy to follow. Maybe a short description of the objective at the beginning could help the reader. What are the objects to be clustered, following which criteria? If I understood well, the goal is to clusters grid-points depending on their temporal correlation between each other for each variables so that grid-points with strong correlation are group together. |

| | |
|---|---|
| AC | Agreed. We will rephrase the opening sentence of the section to:

"With the spatial networks, the graphs, from Sect.~3.2 at hand, we aimed to cluster geographical points (represented as nodes in each network), so that areas with similar temporal behaviour are grouped together." |
| | |
| | Results and Discussion |
| RC55 | Sec. 4.1: As mentioned before, mentioning which length scale (the one from sec. 3.2.1 or sec. 3.2.3) the authors refer to would help the reader. Since two definition of length scale seems to be used, it feels natural to wonder how they compare? |
| AC | Agreed, we will open section 4.1 with:
"Figure 4 shows the estimated correlation lengthscales for each model variable using three correlation thresholds (0.5, 0.6 and 0.7) as found from the analysis described in Sect. 3.2.1."

This makes it clear which length scale is being spoken about. |
| | |
| RC56 | l. 275-278: I think I got the general idea here: the spatial distribution of the length scale of a specific variable is the product between Fig. 5a and Fig. 4. However, as it seems that it is not the same definition of the length scale between Fig. 4 and Fig. 5a it is a bit confusing. |
| AC | Yes, we will make this clearer as to which definition is being used. |
| | |
| RC57 | Sec. 4.1: I am not familiar with length scale, but it seems that there is some literature on length scales (just saying that based on a quick search on google scholar). Some comparison of the results and the methods with the literature is missing there. Are there other definition of length scale? How does the method used in this paper compare with other? Are the length scales similar to former estimations? |
| AC | In variational DA where we often parametrize the horizontal length-correlations, the lengthscales can be supplied as a free parameter fitting a specific function (e.g. Gaussian, Lorenzian, SOAR, Gaspari-Cohn) The length-correlation functions are identified either by ensemble runs, or diagnostic methods, such as by Desroziers et al. (2005). For example, the UK Met Office system on the NWES uses sum of two Gaussian functions and the length-correlation functions have been recently re-assessed through diagnostic methods by Fowler et al (2022). The ambition of this work is not to provide fitted functions for the length-correlation, but rather assess through single length-scale parameter how the length-correlations spatially vary across the NWES. This can then feed into future length-correlation analysis. We will compare the spatial length-scale maps from this manuscript with the only study (we are aware of) that analysed biogeochemistry length-correlations on the NWES in some detail, the Fowler et al (2022) paper, which is already cited in our manuscript. We will include discussion on this comparison in the upcoming revision of our manuscript. Please note that as stated in our response to reviewer's comment 2 (RC41), we will also add to the list of references other, more general, papers on the length-scale estimation. |
| | |
| RC58 | Fig. 7: How is it done? I guess it is some kind of generalisation of Fig. 6 but it would be good to know more than « We used those robust boundaries to identify 13 regions representing areas of NWES connectivity. Results of this regionalisation are represented in Fig. 7. » (line 315) |
| AC | Yes, it is a generalisation of Fig 6. We will expand the caption for Fig. 7 to give more detail. |
| | |

| | |
|---|---|
| RC59 | l. 350: « or build simpler models than ERSEM » I think this need to be say a bit differently. Complexity of models tends to increase to better (or hoping to better) represent the real world. NPZD models already exist with just one phytoplankton, one zooplankton... Here the issue is to simplify ERSEM while keeping an accurate representation. Maybe something like line 51 « simplified (yet realistic with respect to the objectives) ». |
| AC | Agreed, we will add the following to improve readability as suggested:
"or build simplified (but realistic with respect to the objectives) models than ERSEM." |
| | |
| RC60 | l. 363-366: I do not see that in Fig. 8. The mean correlation between POM (yellow) and the Higher Trophic Levels + DOM (pink) is rather low. The authors should clarify. |
| AC | Current text, "Fig. 8 demonstrates two more clusters of variables grouped together: the group of particulate organic matter (POM)"
We will remove "grouped together" to clarify the meaning.  These words are not necessary, and they might imply that the clusters are linked in some way. |
| | |
| | |
| | Conclusions |
| RC61 | l. 410-426: You are here a bit more specific about the results and this could be used for the abstract. E.g. « we can conclude that the biogeochemical lengthscales vary significantly between variables and are not directly transferable. » or « we have provided an approximation for the lengthscale of each variable, and each spatial location, that is informed by the high correlation in the spatial variability between lengthscales of each variable »... |
| AC | Agreed. We will add a sentence to the abstract that gives some more detail about the length-scale results. |
| | |
| RC62 | l. 421-424: « Our analysis demonstrated that the chemical components (e.g., nitrogen, carbon, silicon. . . etc) of each pelagic variable (e.g., diatoms, nanophytoplankton, microzooplankton) are closely linked and a simpler version of the model can be built, by reducing these variables through parametrization. » I do not know ERSEM but I assume that as many models it started from a simple version and the complexity has been increased (e.g. addition of more phytoplankton types). I am wondering how the grouping compare with a former simpler version of ERSEM? I suppose it should be relatively similar (e.g. all types of phytoplankton in gather in only one) however it will be quite interesting if some grouping where different. |
| AC | Early version of ERSEM has been published in Baretta et al (1995). It is however only slightly simpler model than the current one, i.e the small and larger phytoplankton functional types are in the older model grouped together. This corresponds well with our clustering. |
| | |
| | Extra comments |
| RC63 | « lengthscales »: After a quick search on google scholar, it seems that it is rather written « length scales » or « length-scales ». |
| AC | We agree that 'lengthscale' is a less common spelling of the word.
To better align with other literature, we will replace instances of 'lengthscale' with 'length-scale'. |
| | |
| RC64 | The regions define in Fig. 7 could be used for sampling the domain to analyse the inter-variable interaction network. Maybe selecting grid points only within one region and to compare with the same done with another region. Are the interaction between variables different between two regions? Or sampling evenly between the regions to have a fair |

| | |
|---|---|
| | general representation? This point is mostly for curiosity as it seems natural to try to use these regions. |
| AC | This is interesting and would well worth considering in future work. However, it is out-of-scope for this particular work. |
| | |
| RC65 | l. 367: Butenschon et al. (2015) and Butenschon et al. (2016) are similar paper (2015 is the discussion version of 2016). Better to keep only 2016. |
| AC | Agreed, we will change accordingly. |
| | |

Let us thank again the reviewer for their important suggestions and we hope that after the suggested changes, addressing the reviewer comments, the manuscript will be in a good shape to be accepted for publication.

Best wishes,

Ieuan Higgs and the co-authors

---

## Author Response (AR1)

Thank you for taking the time and care to provide valuable feedback and contributions to this manuscript. We have updated our manuscript according to the changes, which has strengthened the work.

Copy of review comments (RC) are given below, followed by the author comments (AC).

**Responses to RC1:**

| RC00 | This paper analyses output from a complex biogeochemical model, ERSEM, using network analysis. The analysis is used for several purposes: evaluating the spatial length scale of the variables, determining areas of coherent biogeochemical interactions and boundaries of low connectivity, and establishing which variables are highly connected with each other. This information is useful when setting up regional systems. and evaluating the interactions between model variables and weather the system can be approximated well by a simpler representation. The length scales are useful in data assimilation systems, when setting the area of influence of the observations. I think the paper provide new knowledge worth publishing, but before I would like the following points addressed: |
|---|---|
| RC01 | - Only surface data is used, this is reasonable to reduce the amount of data, but it would require a discussion of the implications of such a choice. For example, in the resulting network from the analysis (Figure 9) the detritus is completely disconnected from the photo and zooplankton, but as that quickly sinks out it would not remain one on the surface and maybe using only surface data is the reason for this disconnect? There is also a question wether there are other methods to reduce the data size that would retain more information throughout the water-column that could have been used? |
| AC | This has been addressed with the addition of text discussing the limitations/implications on Line 82. |
| | |
| RC02 | - The longer time-scales are filtered out, so there could be biogeochemical feedback mechanisms that work on timescales >10 days that are filtered out. So what happens when resulting network is used to inform an emulator, and then applied in the context of climate as suggested by the authors? This also needs to be addressed in the discussion. |
| AC | We have added text to address this and justify our choice on Line 138 onwards. |
| | |
| RC03 | - Applicability of results: Would this results of the analysis be valid other models? For example could the length scales obtained be used in data assimilation system using another BGC model than ERSEM? Would the length scales apply when assimilation observations deeper in the water column even if your results that are only based on surface model data? |
| AC | The methodology highlighted by this manuscript is naturally applicable to different models and datasets. We would expect the key characteristics derived here from ERSEM to be representative of ecosystem itself, and therefore of other trustworthy representations of the ecosystem (i.e. trustworthy models). Line 324 has been updated to reflect this. Wherever the derived characteristics could be compared to the general knowledge, they compared very well. We note that there are severe constrains on validating our results with observations, due to the intermittency of satellite observations - and doing the same analysis with other models is beyond the scope of this project. This has been discussed in the manuscript, please see the line 76. |
| | We would expect the lengthscales to be broadly applicable in the mixed layer of the ocean, and to be not applicable beneath the mixed layer. The majority of biological growth happens in the mixed layer, and so it is of higher significance and relevance to investigate (we might think differently if we were instead investigating global or deep ocean systems). In addition to this, on the NWES there aren't that many sub-mixed-layer observations, at |

|  | least compared to the number of surface observations, making our knowledge of horizontal lengthscales at these depths more challenging, and not our particular focus. This is reflected in the updated text on Line 448 onwards. |
| --- | --- |
|  |  |
| RC04 | - The description of the methods could be improved for the benefit of the reader, I provide some suggestions for what needs to be clarified below. |
| AC | The suggestions are extremely helpful and go to strengthen the manuscript. Details of each improvement are given with the corresponding suggestion. |
|  |  |

|  | Specific comments |
| --- | --- |
| RC05 | Title: Could the title be improved but adding "Investigating" at the beginning? |
| AC | The title has been updated to "Investigating ecosystem connections in the shelf sea environment using complex networks". |
|  |  |
| RC06 | Abstract:
The expression "functional types of variables" is used in the abstract and in the text, it is a bit unclear to me what this means. The expression becomes particularly confusing since the ERSEM itself also includes functional types of plankton. Consider either using a different expression or define it properly before using it. |
| AC | Different expression has been updated to use "functional groups" when referring to a group of similarly behaving variables, to avoid confusion with PFTs used by ERSEM. See lines 4 & 54. |
|  |  |
| RC07 | "Be also used" should be "also be used" |
| AC | Updated line 5 to correct this. |
|  |  |
| RC08 | What is meant by "flow of information between degree of freedom" |
| AC | Statement was unclear. We have rephrased it to say:
"The clusters indicate geographical regions within which there is a large exchange of information within the ecosystem, while information exchange across the boundaries of these regions is limited." See line 10. |
|  |  |
| RC09 | The first part of the last sentence is unclear to me: I don't see that it is demonstrated anywhere how these results can be used to understand how a perturbation propagate through the ecosystem. |
| AC | We have changed the statement:
"The results of this study describe how information is expected to propagate through the shelf-sea ecosystem, and how it can be used in multiple future applications such as stochastic noise modelling, data assimilation, or machine learning." See line 11. |
|  |  |

| RC10 | Line 38: "…investigate three relevant questions related …" either formulate the three topic as questions or rewrite the sentence on line 38. |
|------|---|
| AC | We have rewritten the sentence on line 39 to read "…investigate three relevant *topics* related …" |
| | |
| RC11 | Line 40: "based on" should be "apply". |
| AC | Updated the text, now on Line 43. |
| | |
| RC12 | Line 40: Is this length scale only useful when applying variational data assimilation, not other (ensemble) data assimilation techniques? |
| | We have updated the text to reflect the relevance to ensemble data assimilation (EnDA) too, on line 46. "Furthermore, such length-scales can also inform localization factors within an ensemble DA method." |
| | |
| RC13 | Line 49: as mentioned before, the use of the expression the use of the expression "functional type" is a bit confusing, please define it here. |
| AC | Rephrased this expression in line with our earlier comment on "functional types" -> functional groups". Here we will also define a functional group (for additional clarity): "functional groups (i.e. a set of state variables that are generally highly correlated with each other)." On line 54. |
| | |
| RC14 | Line 51: The statement that these traditional biogeochemical models are unsuitable to address response to climate change, effectively writing off all CMIP simulations is quite severe, I would suggest to moderate the statement. However I do agree that lighter model systems are more suitable for ensemble simulations, but it they are trained on data from the present day, they may not be very good at representing future ecosystem response. |
| AC | We added text on line 58 to moderate the statement, making it more specific to testing a wide variety of scenarios using lighter weight models, as suggested: "…provide additional information to biogeochemistry modellers for building simplified (yet realistic with respect to the objectives) and computationally cheaper models than ERSEM, capable of simulating wide range of what-if scenarios." |
| | |
| RC15 | Line 88: Were the river nutrients also included and were they also annual? |
| AC | We have included text on line 113 to specify the river inputs more clearly: "Both the physical and biogeochemical models were forced by daily varying river discharge data from \cite{lenhart2010predicting}, and initialized from the CMEMS reanalysis produced at the Met Office (product CMEMS-NWS-QUID-004-011, https://marine.copernicus.eu/ services-portfolio/access-to-products/)." |
| | |
| RC16 | Line 120: the transformation to the time-local standardised form is very well explained, but I wonder what happens in period when standar deviation is low or zero (for example I winter), does and stay finite? |
| AC | It stays finite. We have included some additional examples in the Supplementary Material (S2) to show how the filter affects these signals. |
| | |

| RC17 | Line 120: Would river input influence the network results, for example would there be a stronger connection between the biogeochemistry and salinity in a region of strong river influence.  I.e. would the network presented in figure 9 differ from region from region to region? |
|---|---|
| AC | The river inputs influence the network structure, which is clear from Fig. 7 "Region G" – a region largely defined by the delta of River Elbe (see also the difference in nutrients in supplementary material).

 In the context of Fig. 8, we would expect certain connections to be amplified/dampened if we were to reduce the sample size to a specific region. The figures provided aim to show an overview for the system behaviour, acknowledging these effects with the coefficients of variation shown in Fig.8.  While we acknowledge that there is a great variety of additional questions one can ask and explore, in this work, we don't have scope to address all of these questions and we have to be selective (e.g. we don't specifically address the difference between river delta areas and areas further from the coast). |
| | |
| RC18 | Line 124: I did not see it specified anywhere that data were treated any differently, so could you just simply write that all dat were treated this way? |
| AC | Rewritten to simply state:
 "All data used in this study have been preprocessed using the procedure in Eqs…"
 On line 144. |
| | |
| RC19 | Sections 3.2.1: Biogeochemical length scale estimation: What did you do in regions close to land or the boundary? Did you not compute the length scale or only consider the ocean points?  The same question applies to the method in 3.2.2 |
| AC | In Sec. 3.2.1, we only considered the ocean points away from the boundary for this initial calculation of the average lengthscale of each variable. We added text to specify this on line 174.
 In section 3.2.2, we upscale from a 7km -> 21km grid. We use the arithmetic mean of the relevant points to achieve this upscaling (added on line 174) To account for the boundaries, we consider a 21km grid point to be ocean only if more than half of the relevant 7km points are also ocean. In addition, this provided good results, and the lengthscales at the boundaries are discussed in section 4.1:
 "It is notable that another area of low-connectivity is the open (Atlantic) AMM7 domain boundary regions. This indicates that the boundary conditions of the regional model de-correlate from the rest of the domain…" |
| | |
| RC20 | Difference between method in 3.2.1 and 3.2.2: Am I correct that the difference between 3.2.1 and 3.2.2 is that 3.2.1 is done on a finer grid and uses a different method to compute the length scale?  The coarsening before computing the length scale is primarily used to reduce the amount of data given to the SGC?  Is this correct or are there other resort to compute length-scales twice?  This could be made clear in the manuscript. |
| AC | This conclusion is mostly correct. However, we have made this clearer in the manuscript. Some slight clarifications:
 Section 3.2.1 is used to determine the average lengthscale of each variable.
 Section 3.2.2 is used to determine the average lengthscale of each geographical point (independent of the specific variable). Since Sec. 3.2.1 shows that the average lengthscales are different, we account for this by using the different correlation coefficient thresholds on each variables network, such that each network has the same number of links. This effectively normalises the lengthscales for each variable, and allows us to calculate an average for each point. |

| | |
|---|---|
| | We have made this clearer, with text and some simple equations that have been added on line 196 onwards. |
| | |
| RC21 | Line 153: How was the grid upscaled from 7 to 21 km? |
| AC | Text added to specify that "arithmetic averaging" was used on line 174. |
| | |
| RC22 | Line 154-160 Explanation of pruning: This is very hard to understand, please explain better how this was done. |
| AC | We revised text about pruning to make it clearer. We have removed it from this section, and move it to a more appropriate section of the manuscript (Sec. 3.2.3, line 196 onwards) instead of in the section for initially generating the spatial networks (3.2.2), as it is only a minor detail. |
| | |
| RC23 | LIne 170 from "We took …" and the next sentence mean exactly the same thing. Remove the first sentence (or last, up the author, but I preferred the last). |
| AC | We removed the duplicated sentence (see line 190). |
| | |
| RC24 | Before line 180: This is not easy to understand, could you please try to make this clearer: "This was done by taking the mean lengthscale at each grid point across all variables from the dynamically thresholded spatial networks. In order to assess whether this spatial variation could be well approximated by the mean of these lengthscales, we compared the spatial distribution of lengthscales between each different variable using Pearson's correlation. Here, we would expect to see a high correlation if the structure of the spatially varying lengthscales is consistent. This set of spatially varying lengthscales was then represented as a ratio of the mean." |
| AC | This has been re-written, with the addition of some simple equations to make this section clearer. Line 196 onwards. |
| | |
| RC25 | Line 193: "a links … defined by the Spearman correlation.. " at this point there has been introduces severe spearman correlation, the length scale of the correlation with itself on a 7 km grid, the length-scale om a 21 km grid and the correlations between the length scales of different variables, so which one does this refer to here? |
| AC | We have adjusted the text to make specific reference to the relevant case: "defined in this case by the Spearman's correlation between each node on the 21km grid )" on line 218. |
| | |
| RC26 | Paragraph line 190-200: Please write out the equations on its own line (as on page 7) and give them numbers to benefit the reader. |
| AC | Equations have been given their own lines and reference numbers from line 217 onwards. |
| | |
| RC27 | Line 220: This is difficult to follow: "In order to compare the regionalisation of each variable, we first projected the cluster labels of each node back onto the horizontal plane. Then, we applied an edge detection kernel to identify the boundaries between differently labelled regions, creating a boundary map for each variable (with value 1 at boundary grid points and 0 elsewhere)." Please refer back to the appropriate equation on the previous page (ref. my comment above). |
| AC | We have rewritten this paragraph to make it much clearer, and referenced the relevant equations. See line 249 onwards. |
| | |
| RC28 | Line 235: You calculate the mean adjacency matrix over 300 point randomly selected over the shelf <200 meter and then average that. Then later you say "the boundaries particularly seem to reflect shallower bathymetry (approx. 100 m) than the 200 m depth |

| | | |
|---|---|---|
| | | usually applied to delimit the margins of shelf-seas, including NWES." So why not samle within 100 meters? |
| | AC | We chose to use the 200m bathymetry as it is the standard convention to define shelf-seas, including the NWES (Skakala et al (2022), Huthnance et al (2009), Borges et al (2006). This delimitation corresponds to the continental shelf. The 100m area highlights that a big part of this region is linked to the open North Atlantic, which is of interest. *Skákala, J., Bruggeman, J., Ford, D., Wakelin, S., Akpınar, A., Hull, T., Kaiser, J., Loveday, B.R., O'Dea, E., Williams, C.A. and Ciavatta, S., 2022. The impact of ocean biogeochemistry on physics and its consequences for modelling shelf seas. Ocean Modelling, 172, p.101976.* *Huthnance, J.M., Holt, J.T. and Wakelin, S.L., 2009. Deep ocean exchange with west-European shelf seas. Ocean Science, 5(4), pp.621-634.* *Borges, A.V., Schiettecatte, L.S., Abril, G., Delille, B. and Gazeau, F., 2006. Carbon dioxide in European coastal waters. Estuarine, Coastal and Shelf Science, 70(3), pp.375-387.* Additional references have been added to support this. |
| | | |
| RC29 | | Line 255: Be precise: inclusion of new types of observations *for assimilation* ... |
| AC | | Text added on line 289. |
| | | |
| RC30 | | Line 255: I suggest to remove "profound". |
| AC | | Text removed on line 288. |
| | | |
| RC31 | | Line 265: suggest: "oxygen have different lengthscales ..." |
| AC | | Simplified the text to say that the "assumption is not justified. Line 297. |
| | | |
| RC32 | | Line 379: "... we applied SGC...": did you also test different values of k here? |
| AC | | Yes, we mention this on lines 246. We added text on line 413 to clarify in this later text. |
| | | |
| RC33 | | Line 390: "Ammonium dynamics are relatively more complex than the ones of nitrate." This sentence can be removed. |
| AC | | Removed the text on line 423. |
| | | |
| RC34 | | Figure 9: How was the lines connecting the different variables decided? |
| AC | | Added text to Figure 9 caption: "The highest correlations (top 25\%) of all possible pairwise correlations between variables are shown (grey lines)." |
| | | |
| RC35 | | Line 427: I suggest to use another word than "dismantling". |
| AC | | Text changed to "simplifying" instead. Line 462. |
| | | |
| RC36 | | Concerning the supporting information, this would be easier to understand if the variables plotted were given standard names and the y-axis were supplied with the units. |
| AC | | Updated the plots in the supporting information and re-formated them with standard names, and correct units on the y-axis. |
| | | |

Reviewer 2:

| | |
|---|---|
| RC40 | Overview
In this paper, the authors use complex network theory with outputs from a model simulation of the North-West European Shelf (NWES) to identify 1) spatial correlation length scales of biogeochemical variables, 2) geographical regions with strong spatial correlation within them and weak correlation between them and 3) correlations between biogeochemical variables. Point 1) is achieved by computing the Spearman's correlation coefficient between the time series of the different grid points. For point 2), for each variable, they build a spatial network with the previous coefficient, apply spectral graph clustering to gather grid-points and identify the boundaries of these clusters. Then, they define the regions base on the fraction of variables that have a boundary in each grid point. For point 3), they compute the Spearman's correlation coefficient between the spatial distributions of each variable, build a spatial network with that and use the spectral graph clustering to cluster biogeochemical variables. A first result of this work is to show that complex network theory can be used to identify biogeochemical regions based on spatial correlation or to identify correlation between biogeochemical variables. This is of interest for reducing the complexity of biogeochemical dynamics and for helping the analysis of simulations. The correlation length scales are of interest for data assimilation as it quantify the range of the influence between grid points.
I very much appreciated to read the paper. It is clear and well written. The results are of interest and worth to be published. It presents an interesting way to analyse biogeochemical model outputs. The definition of biogeochemical provinces is particularly interesting as it can help the analysis of models. The methods are clearly explained. I do not have major comments on the paper, but rather a list of minor or specific comments that I think could further improve the paper. The comments that are more important are highlighted in red (see the pdf file attached for colored version). |
| | As a summary of my comments, here are my answers to the review criteria at Biogeosciences. I just selected the relevant questions: |
| RC41 | 1. Do the authors give proper credit to related work and clearly indicate their own new/original contribution? Yes. Maybe a bit of comparison with the literature on correlation length scales could benefit the paper. |
| AC | We agree. We have included some comparison to additional literature on lengthscales. Some examples are given below:
*Fowler, A.M., Skákala, J. and Ford, D., 2023. Validating and improving the uncertainty assumptions for the assimilation of ocean-colour-derived chlorophyll a into a marine biogeochemistry model of the Northwest European Shelf Seas. Quarterly Journal of the Royal Meteorological Society, 149(750), pp.300-324.*

*Desroziers, G., Berre, L., Chapnik, B. and Poli, P., 2005. Diagnosis of observation, background and analysis-error statistics in observation space. Quarterly Journal of the Royal Meteorological Society: A journal of the atmospheric sciences, applied meteorology and physical oceanography, 131(613), pp.3385-3396.*

*Hollingsworth, A. and Lönnberg, P., 1986. The statistical structure of short-range forecast errors as determined from radiosonde data. Part I: The wind field. Tellus A, 38(2), pp.111-136.*
*See line 41.* |
| | |
| RC42 | 2. Does the abstract provide a concise and complete summary? Mostly. It could be improved by more clearly stating the results |
| AC | The abstract has been amended according to the "minor and specific comments" . |
| | |

| | |
|---|---|
| |  |
| | Abstract |
| | I think the results should be more clearly/precisely stated in the abstract. It seemed a bit to vague to me. For example: |
| RC43 | - l. 4: « to identify the functional types », which one are they exactly? |
| AC | Different expression used: "functional groups" when referring to a group of similarly behaving variables, to avoid confusion with PFTs used by ERSEM.
*See line 4, 54.* |
| | |
| RC44 | - l. 6: « identifying the (geographically varying) connectivity lengthscales and the clusters of spatial locations that are connected. » What are the main findings concerning the length scales? What are the different clusters? For the length scales, results that seems particularly interesting is that spatial variability is quite similar between variables, requiring only to scale it using the mean length. |
| AC | We have updated the abstract to give more detail on the lengthscale results.
"We show that the biogeochemical length-scales vary significantly between variables and are not directly transferable. We also find that the spatial distributions of length-scales are similar across each variable, as long as a specific scaling-factor for each variable is taken into account. "
Line 7. |
| | |
| RC45 | - l. 9: « The results of this study help to understand how natural, or antrophogenic, perturbations propagate through the shelf-sea ecosystem », it is difficult to agree with that last sentence since the results where not clearly stated before. After finishing reading paper, I also do not think the results help to understand how perturbations propagate in the ecosystem. The results rather offer a analysis framework to do that. |
| AC | The existing statement is rephrased as:
"The results of this study describe how information is expected to propagate through the shelf-sea ecosystem, and how it can be used in multiple future applications such as stochastic noise modelling, data assimilation, or machine learning."
Line 13. |
| | |
| RC46 | - l. 9: « antrophogenic » -> anthropogenic |
| AC | *Text removed. Line 11.* |
| | |
| | Introduction |
| RC47 | l. 35: « an abstraction that will allow for smarter decision-making when considering data sampling and feature selection for ML. » Not that clear to me how and why abstraction can allow smarter decision-making. |
| AC | We agree that the word abstraction" is inappropriate and misleading here, we have replaced it with the word "information". The key message of this sentence is that identifying connections across the NWES and variables, would indicate which variables and locations are unneeded as input features into ML algorithms. See line 36. |
| | |
| RC48 | l. 37-50: Very nice paragraph clearly stating the objective of the work. |
| AC | *Thanks.* |
| | |
| | Model and Data |

| | |
|---|---|
| RC49 | Sec. 2.1: I think it will be nice to have a bit more details about the configuration. Things like: numerical schemes, diffusion, viscosity, equation of state, what forcings (wind, temperature?). How the simulations are run (spin-up procedure, initialisation...). The reference to the papers should be for further details. The reader should not need to read these papers to get a basic understanding of the configuration. |
| AC | We have given more detail about the configuration used. See line 94. |
| | |
| | |
| | Methodology |
| RC50 | Sec. 3.1: maybe a figure showing the raw and filtered time series in the supplementaries could be useful to illustrate what are the timescale filtered? Or maybe some periodogram? It should probably be stated before (introduction? Or somewhere in the methods?) what are the timescales of interest? And why? Out of curiosity have you tried your analysis with the seasonal signal? |
| AC | We have added several figures comparing the raw and filtered time-series in the Supplementary Material S2.

The time-scales are limited by the resolution of outputs (1 day) on one end, and by the need to remove seasonality on the other end, since seasonality introduces artificial correlations. The links between variables are explored through multi-year simulations, since these we could computationally afford. We however believe that the links between variables and regions that we identified here could be applicable to longer time-scales than the time-scale of the simulation.  Note, some tests were done using the seasonal signal, however as already said this signal results in a very large temporal correlation across the entire domain, obfuscating any detail within the region. |
| | |
| RC51 | l. 154: « to a 21 km spatial resolution » make me wonder if the results are sensitive to the resolution of the model? Longer length scale because of eddy mixing? Or shorter one because of dynamical barrier created by filaments or eddies? This somehow questions also the isotropy assumption. |
| AC | As for the nature of the horizontal lengthscales identified in the manuscript: We expect that physics (eddy-mixing, filaments, eddies, which should be resolved by the 7km model) is important contribution to the length-connections (e.g. increase of Rossby radius in the open ocean), but one needs to keep in mind that for the biogeochemical variables these drivers are intertwined with biogeochemical drivers that will also be reflected in the output (e.g. river delta geography, interaction of sunlight with biology...etc).

To clarify the 7km and 21km resolutions: the model is run at 7km resolution, and only the daily model outputs are upscaled to 21km, because the 21km is the highest resolution we could feasibly use in our complex network analysis for computational reasons. When it comes to the model resolution (7km), obviously using higher resolution than 7 km would improve the physics of the model (e.g. increase of the model resolution to 1.5km is an ongoing stream of work at multiple involved institutions), but for the spatial scales considered by this manuscript (I.e. the NWES-wide analysis)  it is widely accepted that 7 km model resolution provides a good approximation to physics and biology. At the end, the 7 km model is used operationally at the UK Met Office and its outputs are supplied to CMEMS. To summarize, we expect that increasing spatial resolution would lead to only higher order corrections to the results presented in this work. |

| | |
|---|---|
| | |
| RC52 | l. 162: I do not understand why the authors say : « As opposed to the biogeochemical lengthscales computed in Sect. 3.2.1 [...] here we manipulate the spatial networks to look at the spatial dependency of this length scale. » In section 3.2.1 you also have a map of the length scales that give you the spatial information (Fig. 2). I do not get the interest of these two definitions. Note that this also bring a bit of confusion about which are the length scales used for the different plots. For example in Fig. 4 which one is it? And for Fig. 5? I kind of got that Fig. 4 is the length scale define in sec. 3.2.1 and Fig. 5 the one in sec. 3.2.3 but it is not so clear. |
| AC | It is correct that 3.2.1 relates to Fig. 4, and Sec. 3.2.3 relates to Fig.5.
We have added some text to Fig. 2's captions to make clearer – this is just a visualisation of what a length scale calculation looks like.
We have removed and added text that clear up this confusion. See line 174, 177, 196. |
| | |
| RC53 | l. 167: « black » rather than « red »? |
| AC | Correct. Updated line 187. |
| | |
| RC54 | Sec. 3.3: This part is not easy to follow. Maybe a short description of the objective at the beginning could help the reader. What are the objects to be clustered, following which criteria? If I understood well, the goal is to clusters grid-points depending on their temporal correlation between each other for each variables so that grid-points with strong correlation are group together. |
| AC | Agreed. We have updated line 206 onwards to make this clearer at the start of section 3.3 |
| | |
| | Results and Discussion |
| RC55 | Sec. 4.1: As mentioned before, mentioning which length scale (the one from sec. 3.2.1 or sec. 3.2.3) the authors refer to would help the reader. Since two definition of length scale seems to be used, it feels natural to wonder how they compare? |
| AC | Section 4.1 starts with:
"Figure 4 shows the estimated correlation lengthscales for each model variable using three correlation thresholds (0.5, 0.6 and 0.7) as found from the analysis described in Sect. 3.2.1." |
| | |
| RC56 | l. 275-278: I think I got the general idea here: the spatial distribution of the length scale of a specific variable is the product between Fig. 5a and Fig. 4. However, as it seems that it is not the same definition of the length scale between Fig. 4 and Fig. 5a it is a bit confusing. |
| AC | Yes, we have made this clearer. See line 301. |
| | |
| RC57 | Sec. 4.1: I am not familiar with length scale, but it seems that there is some literature on length scales (just saying that based on a quick search on google scholar). Some comparison of the results and the methods with the literature is missing there. Are there other definition of length scale? How does the method used in this paper compare with other? Are the length scales similar to former estimations? |
| AC | In variational DA where we often parametrize the horizontal length-correlations, the lengthscales can be supplied as a free parameter fitting a specific function (e.g. Gaussian, Lorenzian, SOAR, Gaspari-Cohn) The length-correlation functions are identified either by ensemble runs, or diagnostic methods, such as by Desroziers et al. (2005). For example, the UK Met Office system on the NWES uses sum of two Gaussian functions and the |

| | | |
|---|---|---|
| | length-correlation functions have been recently re-assessed through diagnostic methods by Fowler et al (2022). The ambition of this work is not to provide fitted functions for the length-correlation, but rather assess through single length-scale parameter how the length-correlations spatially vary across the NWES. This can then feed into future length-correlation analysis. We will compare the spatial length-scale maps from this manuscript with the only study (we are aware of) that analysed biogeochemistry length-correlations on the NWES in some detail, the Fowler et al (2022) paper, which is already cited in our manuscript. We will include discussion on this comparison in the upcoming revision of our manuscript.

 Please note that as stated in our response to reviewer's comment 2 (RC41), we have also added references for other, more general, papers on the length-scale estimation. | |
| | | |
| RC58 | Fig. 7: How is it done? I guess it is some kind of generalisation of Fig. 6 but it would be good to know more than « We used those robust boundaries to identify 13 regions representing areas of NWES connectivity. Results of this regionalisation are represented in Fig. 7. » (line 315) | |
| AC | Yes, it is a generalisation of Fig 6. We have added more detail to the caption for Fig. 7 to make it clearer. See Fig.7 caption. | |
| | | |
| RC59 | l. 350: « or build simpler models than ERSEM » I think this need to be say a bit differently. Complexity of models tends to increase to better (or hoping to better) represent the real world. NPZD models already exist with just one phytoplankton, one zooplankton... Here the issue is to simplify ERSEM while keeping an accurate representation. Maybe something like line 51 « simplified (yet realistic with respect to the objectives) ». | |
| AC | Agreed, we have added the following to improve readability as suggested:
 "or build simplified (but realistic with respect to the objectives) models than ERSEM."
 See line 383. | |
| | | |
| RC60 | l. 363-366: I do not see that in Fig. 8. The mean correlation between POM (yellow) and the Higher Trophic Levels + DOM (pink) is rather low. The authors should clarify. | |
| AC | Old text, "Fig. 8 demonstrates two more clusters of variables grouped together: the group of particulate organic matter (POM)"
 We have removed "grouped together" to clarify the meaning. These words are not necessary, and they might imply that the clusters are linked in some way. See line 397. | |
| | | |
| | | |
| | Conclusions | |
| RC61 | l. 410-426: You are here a bit more specific about the results and this could be used for the abstract. E.g. « we can conclude that the biogeochemical lengthscales vary significantly between variables and are not directly transferable. » or « we have provided an approximation for the lengthscale of each variable, and each spatial location, that is informed by the high correlation in the spatial variability between lengthscales of each variable »... | |
| AC | Agreed. We have added a sentence to the abstract that gives some more detail about the length-scale results. See line 447. | |
| | | |
| RC62 | l. 421-424: « Our analysis demonstrated that the chemical components (e.g., nitrogen, carbon, silicon. . . etc) of each pelagic variable (e.g., diatoms, nanophytoplankton, microzooplankton) are closely linked and a simpler version of the model can be built, by reducing these variables through parametrization. » I do not know ERSEM but I assume | |

| | |
|---|---|
| | that as many models it started from a simple version and the complexity has been increased (e.g. addition of more phytoplankton types). I am wondering how the grouping compare with a former simpler version of ERSEM? I suppose it should be relatively similar (e.g. all types of phytoplankton in gather in only one) however it will be quite interesting if some grouping where different. |
| AC | Early version of ERSEM has been published in Baretta et al (1995). It is however only slightly simpler model than the current one, i.e the small and larger phytoplankton functional types are in the older model grouped together. This corresponds well with our clustering. |
| | |
| | Extra comments |
| RC63 | « lengthscales »: After a quick search on google scholar, it seems that it is rather written « length scales » or « length-scales ». |
| AC | To better align with other literature, all instances of 'lengthscale' have been replaced with 'length-scale'. This change is visible throughout the entire paper. |
| | |
| RC64 | The regions define in Fig. 7 could be used for sampling the domain to analyse the inter-variable interaction network. Maybe selecting grid points only within one region and to compare with the same done with another region. Are the interaction between variables different between two regions? Or sampling evenly between the regions to have a fair general representation? This point is mostly for curiosity as it seems natural to try to use these regions. |
| AC | This is interesting and would well worth considering in future work. However, it is out-of-scope for this particular work. |
| | |
| RC65 | l. 367: Butenschon et al. (2015) and Butenschon et al. (2016) are similar paper (2015 is the discussion version of 2016). Better to keep only 2016. |
| *AC* | This has been updated. |
| | |

**Additional Changes**

| | |
|---|---|
| AC66 | Figure 4 had been formatted incorrectly in the first submission, meaning some of the bars on the chart were aligned with the wrong label on the x-axis. |
| AC | This has been fixed. See Fig.4. |
| | |

Best wishes,

Ieuan Higgs and the co-authors

---

## Author Response (AR2)

Thank you for taking the continued time and care to provide valuable feedback and contributions to this manuscript. The technical changes have been made and improved the details of the manuscript. Copy of review comments (RC) are given below, followed by the author comments (AC).

**Report 1**

| | |
|---|---|
| RC | Thanks to the authors for their thorough response to my comments. The revised abstract clearly emphasizes the key findings, and the inclusion of references to more general papers on the length scale gives more context to the reader who is not familiar with the subject. The supplementary figure S2 clearly explain well the temporal filtering. |
| RC | The distinction between the two estimations of the length-scale is much clearer, but the advantage of the one based on the spatial network is still not crystal clear to me. Note, however, that correlation length-scales and spatial network are not my expertise. I still have the impression that you can get the spatial information from the first method (sec. 321). It has been applied to 300 points and then average to get a spatial average. Why not applying it to all the grid points and not spatially averaging? In the description of the second methods (l. 184), it is said that the length-scales estimated with the second method are comparable between the variables in each geographical location. I do not understand why (as well as why the first length-scale estimation it is not comparable) but it seems like an important motivation and should probably be stated at the beginning of the paragraph. |
| AC | It would be too computationally costly to apply the method from section 3.2.1 to every point, for every (potentially) desired correlation threshold (this method is somewhat restricted by the need to set the same static correlation threshold across all variables). The advantage of the methods applied in Sect. 3.2.3 (Line 184) is that we do not need to select a correlation threshold (as we do in Sect. 3.2.1) across each variable's network, and instead we dynamically adjust the correlation threshold for each network so that every network, $N(x, v)$, has the same number of links (an effective way of normalising the relational data). This makes the method from 3.2.3 much more computationally cheap than the method in 3.2.1. Lines 199-204 describe how this is useful in calculating $M(x)$ – a variable independent length-scale map which is representative of the normalised maps $N(x, v)$ if the spatial pattern exhibited by all of the variables is similar.

 As suggested by the reviewer, to clarify for future readers why we cannot use the method from 3.2.1 in section 3.2.3, we added new text on line 179 onwards:
 *"Deducing the length-scales from the networks allows us to bypass the high computational cost associated with the method based on calculating spatial correlations along circles with increasing radii, described in Sect. 3.2.1.*
 *The network-based analysis from this section also provides a convenient way to normalise the structure of the spatial patterns found in each variable, without the need to select a specific correlation threshold as in Sect. 3.2.1."* |
| | |
| RC | Few extra comments:
 l. 125: isn't that a running average with a 10 days time window? If yes, maybe you can use this term. |
| AC | Agreed – added this term. |
| | |
| RC | l. 303: close the parenthesis after « according to Sect. 3.2.3. » |
| AC | Agreed. |
| | |

**Report 2**

| | |
|---|---|
| RC | As it is the second time I review this paper, I think my initial concerns have been addressed and the paper can be accepted for publication. A few very minor issues has been included below for some last polishing on the manuscript before publication. |
| RC | Abstract:
The sentence "We also find that the spatial distribution of length-scales are similar across each variable, as long as a specific scaling-factor for each variable is taken into account." Was hard to understand without reading the entire paper, I propose based on my understanding:
On my initial reading of the abstract, this sentence puzzled me, now after reading though the manuscript, it is still not clear to me what is meant, consider rephrasing, perhaps use "spatial pattern instead of "spatial distribution". |
| AC | Agreed. Changed accordingly. |
| | |
| RC | Suggest: "boundaries of these"- > "boundaries between these" |
| | Agreed. Changed accordingly. |
| | |
| RC | Line 261: suggest "As with before…" |
| AC | Agreed the sentence could be made clearer. "As with before…" doesn't exactly fit, as this section is approaching the data differently.
Re-written to say:
*"The work in the previous sections focused on understanding how each variable separately behaves in horizontal space. In this section, we focused on developing an…"* |
| | |

Best wishes,
Ieuan Higgs and the co-authors